# Variation Matters: from Mitigating to Embracing Zero-Shot NAS Ranking Function Variation

**Pavel Rumiantsev**                                                       *pavel.rumiantsev@mail.mcgill.ca*
*The Department of Electrical and Computer Engineering*
*McGill University*

**Mark Coates**                                                                    *mark.coates@mcgill.ca*
*The Department of Electrical and Computer Engineering*
*McGill University*

**Reviewed on OpenReview:** *https://openreview.net/forum?id=SbGt9Odxdp*

## Abstract

Neural Architecture Search (NAS) is a powerful automatic alternative to manual design of a neural network. In the zero-shot version, a fast ranking function is used to compare architectures without training them. The outputs of the ranking functions often vary significantly due to different sources of randomness, including the evaluated architecture's weights' initialization or the batch of data used for calculations. A common approach to addressing the variation is to average a ranking function output over several evaluations. We propose taking into account the variation in a different manner, by viewing the ranking function output as a random variable representing a proxy performance metric. During the search process, we strive to construct a stochastic ordering of the performance metrics to determine the best architecture. Our experiments show that the proposed stochastic ordering can effectively boost performance of a search on standard benchmark search spaces.

## 1 Introduction

Zero-shot Neural Architecture Search (NAS) is a way to perform selection of a neural architecture without training every candidate architecture. Recently, zero-shot NAS has been applied to a variety of practical tasks, including medical image processing (Wang et al., 2024), sequence processing (Serianni & Kalita, 2023), language model architecture enhancements (Javaheripi et al., 2022) and general image classification (Jiang et al., 2024; Li et al., 2022).

In general, a zero-shot approach consists of two parts: a ranking function and a search algorithm (Xu et al., 2021; Rumiantsev & Coates, 2023; Cavagnero et al., 2023a; Li et al., 2022; Jiang et al., 2024). The ranking function aims to capture the quality of a particular candidate architecture in the form of a scalar value, which allows two architectures to be easily compared (Abdelfattah et al., 2021). A search algorithm uses a ranking function to determine the best architecture over the given *architectural search space*. For this work, we view a search space as a combination of a feasible architecture set and a dataset of labelled training, validation, and test samples.

While the zero-shot approach is the fastest way to find an architecture that performs well, it suffers from variations in the output of the ranking function. In order to assess an architecture as quickly as possible, most zero-shot techniques sample a single batch of data, and randomly initialize the architecture. The ranking function then uses the batch of data and the initialized network to derive a scalar score. The batch selection and initialization thus induce considerable randomness in the output. There are several data-agnostic ranking functions, including SynFlow (Abdelfattah et al., 2021) and LogSynFlow (Cavagnero et al., 2023a), which use a batch of *constant unit inputs*, and Zen-Score (Lin et al., 2021), which uses a batch of *randomly generated* data as an input. Thus, there is still randomness, and the same problem remains.

Typically, zero-shot techniques address this issue by averaging the outputs of the ranking function over multiple batches of data.

A zero-shot search procedure can counter some of the variation of the preferred ranking function by employing a robust search algorithm. For example, Chen et al. (2021) use prune-based search to increase the stability of the search. In general, the performance of a search algorithm can vary considerably, depending on which ranking function is paired with it (Akhauri et al., 2022; Rumiantsev & Coates, 2023). Since it is very challenging to meaningfully analyse a ranking function in isolation (independently of a search algorithm), we base our experimental conclusions on pairs of ranking function and search algorithm. We focus on the most popular ranking functions and search algorithms. Specifically, we select Eigenvalue score (Mellor et al., 2020), ReLU Hamming distance (Mellor et al., 2021) and condition number of the neural tangent kernel (NTK) (Chen et al., 2021) to demonstrate how random search (Mellor et al., 2020) and evolutionary search (REA (Real et al., 2019) and FreeREA (Cavagnero et al., 2023a)) achieve better performance in the majority of cases (>80 percent), when enhanced with our statistical evaluation approach, for standard architectural search spaces including NAS-Bench-101, NAS-Bench-201, and TransNAS-Bench-101.

In this paper, we assess the variation of different ranking functions, explore how common ways to minimize the ranking function variation help to increase the zero-shot performance, and introduce a way to incorporate variation into the assessment of an architecture. Our main contributions are:

- We introduce a ranking function variance analysis framework for zero-shot NAS researchers to analyse existing and newly designed ranking functions.

- We use the variance of a ranking function to statistically compare architectures, and show that this enhances the performance of random and evolutionary search algorithms for zero-shot NAS for a variety of commonly-used ranking functions and architectural spaces.

## 2 Background

### 2.1 Search Algorithms

There are three common choices of search algorithms that are used in Zero-Shot NAS: random search, prune-based search and evolutionary search. **Random search** is the search algorithm with the least computation overhead. It randomly samples a subset of architectures from the search space, and then selects the one with the highest rank according to the chosen ranking function. The size of the sampled subset is the main hyperparameter that controls the trade-off between search speed and efficacy. Due to its simplicity, it is a common choice (Lopes et al., 2021; Xu et al., 2021; Mellor et al., 2020; 2021; Mok et al., 2022).

**Evolutionary search** procedures are based on genetic algorithms. They start with randomly selected architectures, then generate mutations, and then select a subset that are ranked the highest according to the ranking function output. The procedure is then iterated. Mutations can be based on replacement of architectural operators or involve direct manipulations of the architecture encoding. Evolutionary search is slower and more challenging to implement than random search, but the results it produces are more stable. Variations of this search technique are employed more often in recent works (Lin et al., 2021; Akhauri et al., 2022; Li et al., 2022; Mok et al., 2022; Cavagnero et al., 2023a;b; Yang et al., 2023; Wang et al., 2024). Often, the ranking function is applied to multiple batches of data, with the results averaged, in order to provide a more stable score. This procedure is usually performed once, when the architecture is first encountered during the search, with the calculated score cached for later use.

The third type of search algorithm is **prune-based search**. It iteratively removes edges from the hypergraph that can be used to represent all architectures. By estimating the difference in ranking function output for the architectures represented by the hypergraph with and without a given edge, the algorithm can choose the best edge to prune. This process is repeated until the hypergraph represents a single architecture. This type of search introduces significant computational overhead as it requires to evaluate the entire hypergraph for every unpruned edge (Chen et al., 2021). It is the most challenging to implement and, therefore, is rarely adopted (Chen et al., 2021; Rumiantsev & Coates, 2023; Jiang et al., 2024).

## 2.2 Zero-shot ranking functions

Many zero-shot ranking functions have been proposed, and we now review some of the most prominent. Abdelfattah et al. (2021) repurposed metrics previously employed in neural network pruning. The saliency metrics `Snip` (Lee et al., 2019), `Grasp` (Wang et al., 2020), `Fisher` (Turner et al., 2020) and `SynFlow` (Tanaka et al., 2020) measure the impact of removing a specific parameter or operation from the network. In adapting these to construct a zero-shot ranking function, Abdelfattah et al. (2021) sum over all of the architectural parameters, and thus construct a measure of the fitness of the entire architecture. Cavagnero et al. (2023a) introduced a logarithmic version of `SynFlow` named `LogSynFlow`. Applying a logarithm suppresses some of the gradient explosions that can unduly influence the score.

The first two ranking functions designed specifically for the zero-shot setting were the `Eigenvalue score` (Mellor et al., 2020) and the Hamming distance between the ReLU regions (Mellor et al., 2021). Based on the Neural Tangent Kernel (NTK) (Jacot et al., 2018), the `Eigenvalue score` uses the eigenvalues of the gradient correlation matrix to predict the learning capabilities of an architecture. The `ReLU Hamming distance` is constructed as a determinant of the kernel of Hamming distances betweent activation patterns of ReLU, where a pattern is represented as a string of zeros and ones. An alternative measure is the number of activated ReLU regions for a sampled batch of data (Xiong et al., 2020). The underlying conjecture is that activation of more regions allows the network to learn more complex data dependencies. Similarly, Peng et al. (2024) proposes to count the number of unique ReLU activation patterns across the data dimension. The `Entropic Score` (Cavagnero et al., 2023b) is also based on ReLU activations.

Several other ranking functions are based on the NTK, including (i) the mean value of the NTK (Xu et al., 2021); (ii) the Frobenius norm of the NTK (Xu et al., 2021); and (iii) the negative condition number of the NTK (Chen et al., 2021). The NTK can introduce undesirable computational overhead, so other researchers have explored alternative metrics. Rumiantsev & Coates (2023) show that the Frobenius norm of the Neural Network Gaussian Process (NNGP) kernel can act as an effective ranking function. Jiang et al. (2024) focus on the Pearson correlation kernels of the internal representations of an architecture, and compute the sum of the smallest eigenvalues of these kernels. For all of these proposed ranking functions, the required kernels are estimated using a random batch of data.

Since most benchmark search spaces focus on the classification task, some ranking functions incorporate the available label information. Label-Gradient Alignment (`LGA`) (Mok et al., 2022) constructs a metric based on both the class similarity matrix and the NTK. Lopes et al. (2021) propose `EPE-NAS`, which uses the gradient correlation matrix, and is thus similar to the `Eigenvalue score`, but is computed class-wise. For practical reasons, both `LGA` and `EPE-NAS` are computed using a random batch of data.

Some ranking functions are based on the gradients of the trainable weights. `SweetGrad` counts the number of gradient norms in a predefined interval (Yang et al., 2023). Li et al. (2022) sum the coefficients of variations associated with the weight gradients to compute `ZiCo`.

## 3 Problem statement

A neural architecture search space can be represented as a set of feasible architecutres $\{arch_i\}_{[i=1..N_{SS}]}$ of size $N_{SS}$. Associated with the architecture space is a dataset $\mathcal{D}$ consisting of pairs of features $x \in \mathcal{X}$ and labels $y \in \mathcal{Y}$. In this work we consider computer vision tasks where an architecture $arch_i$ is associated with a final accuracy after training, labelled $acc_i$. A zero-shot ranking function assigns a scalar score to each architecture: $s_i = r(arch_i, \mathcal{D})$. A search is then conducted using these scalar scores, with the goal of selecting the most accurate architecture.

As discussed above, almost every zero-shot ranking function has an element that is evaluated over a batch of input data (e.g., the NTK, gradient coefficients of variations, or logit similarity matrix). To improve the stability of the output, most ranking functions are averaged over multiple random batches. Our goal is to use the multiple ranking function outputs in a more effective way.

In subsequent sections, we explore how sensitive different ranking functions are to the random selection of the input data used in their calculation. We point out how commonly used statistical methods allow us to overcome such sensitivity in order to improve the performance of a zero-shot approach.

## 4 Methodology

This section describes our study into the variability of ranking functions. First, we introduce a variation metric for assessing the variability of a ranking function for a particular search space and batch size. Second, we present our zero-shot search setup and explain how the variation metric can be used to improve performance.

### 4.1 Ranking function variation

It is common for ranking functions to use a real training dataset. In order to assign a score to an architecture, a batch of data is randomly drawn from the dataset. The ranking function scores vary from batch to batch, with the extent of the variation being potentially dependent on the batch size, the dataset, and the task.

We measure ranking function variation over the entire search space by computing, for each architecture, an empirical coefficient of variation for the ranking function output over randomly sampled batches of input data. We then average over the architectures. To ensure the coefficient of variation is an appropriate metric, we have specified a "meaningful zero" for the ranking functions. We discuss this in detail in Appendix G

Denoting the batch size by $B$ and the number of sampled batches by $V$, and recalling that there are $N_{SS}$ architectures in the search space, we define the ranking function variation as:

$$Var_{SS}(r, B, V) \triangleq \frac{1}{N_{SS}} \sum_{i=1}^{N_{SS}} CV\left(\mathcal{M}_i(B, V)\right) \quad \text{for} \quad \mathcal{M}_i(B, V) = \{r(arch_i, d_v)\}_{v=1}^{V}, \tag{1}$$

where $r$ is a ranking function, $d_v$ is a batch of data of size $B$, uniformly sampled from the dataset associated with the search space $SS$, and $CV(\mathcal{X}) = \frac{Var(\mathcal{X})}{Mean(\mathcal{X})}$. A higher average coefficient of variation indicates greater variability with respect to the selection of input data.

Typically, if we know that a ranking function is highly variable with respect to the input data, then this motivates us to enhance the robustness by averaging over multiple random batches. On the other hand, if a ranking function exhibits less variability, then a score can be taken from a reduced number of randomly drawn batches. This leads to a faster search, because ranking function evaluation tends to be the most computationally expensive element of zero-shot NAS.

### 4.2 Search algorithm

Zero-shot architecture search uses a ranking function as a proxy to evaluate the quality of an architecture. In order to compare two architectures, a selected ranking function is evaluated on both of them and the outputs of the ranking function are compared. Assuming positive correlation between accuracy and the adopted metric, the architecture with the higher corresponding metric is considered to be better. The ranking function outputs are subject to randomness, due, for example, to random architecture initialization or random selected input batch. A common technique to mitigate randomness is averaging: a ranking function is evaluated multiple times on the same architecture and the output is averaged over the evaluations.

Instead of averaging, the key component of our approach is a statistical comparison of the architectures. We treat multiple evaluations of the ranking metric as multiple observations of a random variable. We then test for *stochastic dominance*. Viewing the ranking function metrics for two compared architectures as random variables, $X$ and $Y$, we test whether $p(X > k) \geq p(Y > k)$ for all $k$ (and $p(X > k) > p(Y > k)$ for some $k$). This test differs from comparing the mean performance, $\bar{X} > \bar{Y}$. Stochastic dominance implies $\bar{X} > \bar{Y}$, but the converse is not true. Our testing process thus targets a stricter performance discrepancy. This makes the search more effective and efficient, because we navigate towards architectures that are more likely to be genuinely superior.

To perform the comparison, we use the U-value from the Mann-Whitney U-test (Mann & Whitney, 1947). This statistical test assesses the null hypothesis that both sets of observations are drawn from the same distribution. The alternative hypothesis (for a one-sided test) is that the random variable X is stochastically greater than the random variable Y. By comparing the Mann-Whitney U values for the two samples, we can identify one as a superior architecture. We declare such superiority only if the null hypothesis can be rejected at a predetermined significance level (e.g., 0.05). If this is not the case, we cannot meaningfully differentiate between the architectures in terms of predicted performance.

We summarise this approach as a statistical maximum and statistical top-k (see Algorithm 1). Those procedures iteratively search for a stochastically dominating maximum over the given sets of ranking functions outputs. Given that we are checking for the stochastically dominating maximum, the output of our procedure is not permutation invariant in general. As the first example, let us consider two sets of scores, A and B, with A stochastically dominating B. The output of Algorithm 1 will be A for both inputs [A,B] and [B,A]. As a second example, let us consider two sets of scores, A and B, where neither stochastic dominates the other. For input [A, B], the output will be A. For input [B, A], the output will be B. Practically, this is equivalent to a random tie breaker. As random tie breakers are common in search functions, we do not view permutation invariance as a necessary requirement.

Pairwise search is not the only viable design option For example, Dunnett's test could be used to perform multiple comparisons.

---

**Algorithm 1** Statistical MAX and TOP-K pseudocode

---

**Ensure:** Each $r_i$ is a list of ranking function evaluations
    **function** STAT-MAX($r_1, \ldots, r_T$)
        $max \leftarrow r_1$
        **for** $i \leftarrow 2, T$ **do**
            /* Mann-Whitney U-test */
            $p \leftarrow \text{MANNWHITNEYU}(max, r_i)$
            /* $Threshold$ is a significance level */
            **if** $p < Threshold$ **then**
                $max \leftarrow r_i$
            **end if**
        **end for**
        **return** $max$
    **end function**

    **function** STAT-TOPK($k, r_1, \ldots, r_T$)
        $top \leftarrow \emptyset$
        $rs \leftarrow r_1, \ldots, r_T$
        **for** $i \leftarrow 1, k$ **do**
            $m \leftarrow \text{STAT-MAX}(rs_1, rs_2, \ldots)$
            $top \leftarrow top \cup \{m\}$
            $rs \leftarrow rs \setminus \{m\}$
        **end for**
        **return** $top$
    **end function**

---

However, our experience is that search algorithms based on multiple comparison tests are much more computationally expensive. The statistical tests also tend to impose stronger assumptions. Thus, we use pairwise search to minimise computational overhead and extend the applicability of the procedure.

In our work, we use two popular search algorithms: *random* search and *evolutionary* search. Both algorithms are easy to implement for an arbitrary search space and require few additional considerations when paired with a ranking function in a zero-shot setting. For **random search** (Mellor et al., 2020), we sample $N$ architectures randomly from the search space. Then we evaluate a ranking function on each sampled architecture $V$ times. We use the Mann-Whitney U-test to rank the candidates. In the case of ties, we select an architecture randomly from the tied candidates.

The REA (Real et al., 2019) **evolutionary search** starts by sampling a *population* of $N$ architectures randomly from the search space. During each iteration, a subset of $n$ exemplars (architectures) is randomly sampled from the population. We order the exemplars using the Mann-Whitney U-test, with random tie-breaking. The exemplar that is first in this ordering is selected and mutated. In our experiments, we use an operator replacement mutation, so that the original exemplar and the mutated exemplar have an architectural edit distance of one. We evaluate a ranking function on the mutant $V$ times and include the mutated architecture in the population, replacing the oldest ranked exemplar. After $T$ iterations of this procedure, we use the Mann-Whitney U-test to order the population and select the highest ranked.

The FreeREA (Cavagnero et al., 2023a) evolutionary search procedure introduces a small modification to the REA algorithm. After a subset of $n$ exemplars is randomly sampled, two exemplars with the statistically

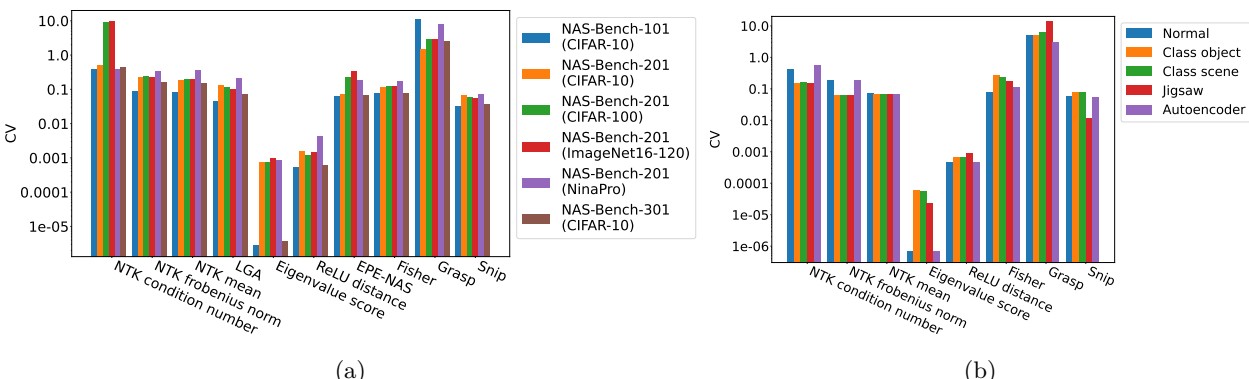

Figure 1: Ranking functions variation $Var_{SS}(r, B, V)$ for ranking function $r$, batch size $B = 64$, $V=10$ and search space $SS$) on (a) NAS-Bench and (b) TransNAS-Bench-101 demonstrate that different ranking functions have considerably different variations within the search space, but a specific ranking function's variation remains similar for the different search spaces.

highest rank are selected. Both exemplars are mutated and the two mutants are included in the population. *Crossover* is performed by splicing together parts of the two selected exemplars. Both mutated exemplars and the resultant crossover exemplar are included into the population replacing the three oldest exemplars, which increases the exploration capabilities of the search process. We provide pseudocode for the evolutionary algorithms in Appendix B.

## 5 Experiments

### 5.1 Ranking function variation

In the first experiment, we show how different ranking functions are influenced by a randomly chosen batch of data and examine how consistent this behaviour is over a variety of search spaces. A lower coefficient of variation means that a ranking function is generally more reliable, can provide more consistent output, and requires fewer batches.

We measure the variation $Var_{SS}(r, B, V)$ by evaluating equation 1. Recall that $SS$ denotes the search space, $r$ denotes the ranking function, $B$ is the batch size, and $V$ is the number of evaluations. In this experiment, in order to focus on the most influential source of variability – the input batch – we initialize each network once, before the first batch is selected. We set $V = 10$ and set $B = 64$. An output of a ranking function is averaged over 10 runs for different random batches given the same initialisation for the network.

We present results for the NAS-Bench search spaces in Fig. 1a and results for the TransNAS search spaces in Fig. 1b. Overall, Eigenvalue score and ReLU Hamming distance are by far the most stable ranking functions. Snip is more stable than the others. Fisher exhibits stability similar to that of Snip on the NAS-Bench search spaces, which are considered easy, but performs worse on TransNAS-Bench. The performance of Fisher and NTK-based kernels are highly dependent on the dataset used by a search space. We emphasize that low variation is a desirable criterion for the ranking function selection, but it does not imply high performance.

For all ranking functions, the coefficient of variation of an architecture tends to decrease as the average accuracy increases, although this effect is more pronounced for some ranking function and search space combinations. In general, this is a result of the ranking function output varying less for the higher quality architectures. Fig. 2 presents example results for ReLU Hamming distance on NAS-Bench-101 (CIFAR-10) and Eigenvalue Score on NAS-Bench-201 (ImageNet16-120). Due to the very high variability often exhibited by poorly performing architectures, we exclude the worst-performing decile from the analysis, and present box plots of the coefficients of variation (for individual architectures) versus accuracy on the test set. We observe a negative trend in each case, although it is much more evident for Eigenvalue Score.

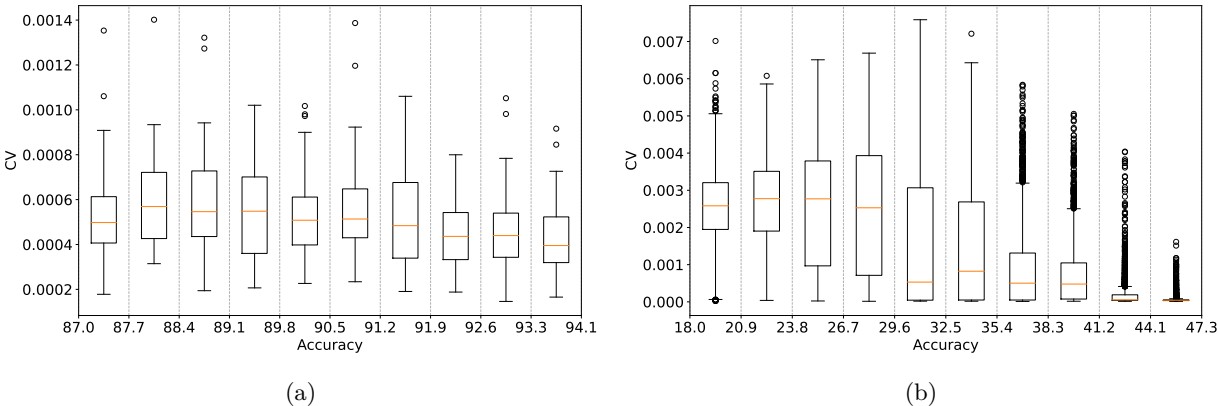

Figure 2: Box plot (versus accuracy) of coefficients of variation for individual architectures ($CV(\mathcal{M}_i(B,V)) = \frac{Var(\mathcal{M}_i(B,V))}{Mean(\mathcal{M}_i(B,V))}$ where $\mathcal{M}_i(B,V) = \{r(arch_i, d_v)\}_{v=1}^{V}$ is a set of $V$=10 ranking function outcomes for architecture $arch_i$ derived from random batches of data $d_k$, each of size $B = 64$. The depicted ranking functions are ReLU Hamming distance on NAS-Bench-101 (CIFAR-10) (a) and Eigenvalue Score on NAS-Bench-201 (ImageNet16-120) (b). For visualization purposes, we exclude ten percent of the architectures-those that exhibit lowest accuracy.

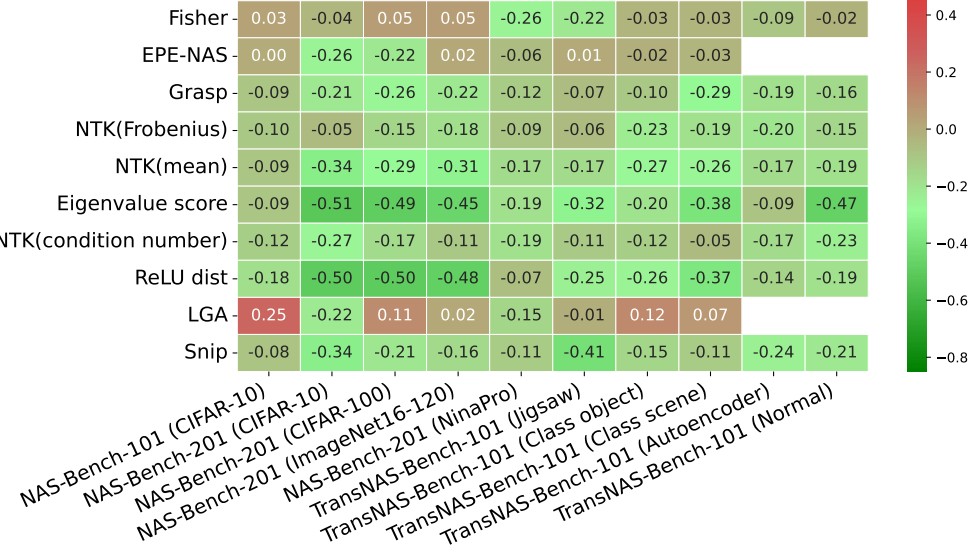

Figure 3: Kendall-$\tau$ correlation coefficient between coefficient of variation values and validation accuracy, for each ranking function and search space. EPE-NAS and LGA require classification labels and therefore cannot be computed for TransNAS-Bench-101 (Autoencoder) and TransNAS-Bench-101 (Normal).

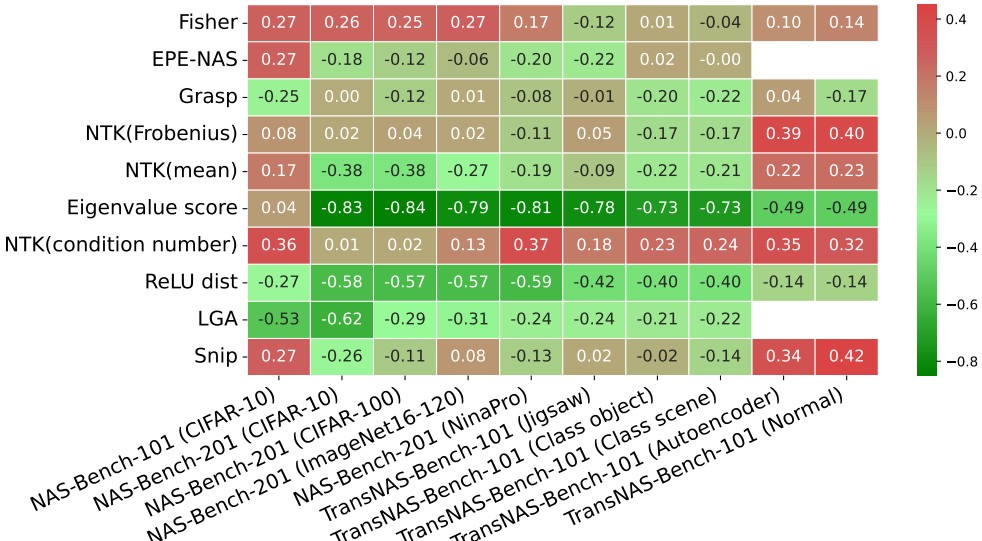

Figure 4: Kendall-$\tau$ correlation coefficient between a ranking function coefficent of variation values and ranking function mean value taken over 10 evaluations, for each ranking function and search space. EPE-NAS and LGA require classification labels and therefore can not be computed for TransNAS-Bench-101 (Autoencoder) and TransNAS-Bench-101 (Normal).

The negative trend suggests that we could consider directly using lower coefficient of variation as an indicative variable during the search. To further investigate this possibility, we measured Kendall-$\tau$ correlation between ranking function variation and architecture accuracy (see Fig. 3). In many cases, there is a strong negative correlation, but there are exceptions. For example, there is very weak correlation between accuracy and the coefficient of variation of the Fisher metric. On some search spaces, LGA even exhibits a positive correlation, meaning that it is, on average, more unstable for the better-performing architectures. We conjecture that this behaviour arises due to the use of labels in both the LGA ranking function and the accuracy evaluation. EPE-NAS, the other ranking function that uses labels, exhibits close-to-zero (and occasionally slightly positive) correlation.

We can consider naive approaches to incorporate the coefficient of variation into the search process. For example, after appropriate normalization, we could sum a ranking function output with its coefficient of variation. Fig. 4 depicts the Kendall-$\tau$ correlation between the coefficient of variation of a ranking function and its mean value for different search spaces. For some ranking functions such as Eigenvalue score and ReLU distance, the magnitude of the correlation is high, meaning that the coefficient of variation does not provide much additional useful information for the search. By contrast, for other ranking functions such as Frobenius norm of the NTK and the condition number of NTK, the magnitude of the correlation is low. In experiments (see Table 6 in Appendix D) we observe that the search performance improvement of our approach over this naive scheme is indeed considerably greater for these ranking functions. Although a naive incorporation of the coefficient of variation can improve results of the search, its impact is inconsistent; the statistical comparison approach we propose in Section 4.2 exhibits more consistent improvement and is effective for all ranking functions.

## 5.2 Random search

In order to demonstrate our statistical approach, we first use a random search proposed by Mellor et al. (2020). Results are presented in Table 1. We selected Eigenvalue score, ReLU Hamming distance and condition number of NTK for this experiment. The first two serve as examples of ranking functions with low variation, while the final one exhibits high variation. To evaluate an architecture we randomly sample a batch of 64 and randomly initialise architecture weights. This evaluation simulates a typical case of

Table 1: Random search on $N$ sampled architectures with averaging (Avg) and statistic (Stat) evaluation. Mann-Whitney U-test is used for statistical evaluation. Each ranking function is evaluated 10 times for the architecture. We provide mean values and standard deviations over 100 runs. For each pair of evaluations, the best score is highlighted in **bold**; if the higher score is statistically significant with p-value below 0.05, it is highlighted in **bold underlined**.

| Search Space | Eigenvalue score | | ReLU dist | | NTK(cond) | |
|---|---|---|---|---|---|---|
| | Avg | Stat | Avg | Stat | Avg | Stat |
| **N= 10** | | | | | | |
| NAS-Bench-101 (CIFAR-10) | 91.51±1.44 | **91.71±2.22** | 91.28±1.80 | **91.63±1.51** | 90.94±1.42 | **91.24±1.63** |
| NAS-Bench-201 (CIFAR-10) | 91.33±1.57 | **92.21±1.66** | 91.80±1.33 | **__92.34±1.14__** | 91.61±1.96 | **91.92±2.15** |
| NAS-Bench-201 (CIFAR-100) | 68.61±3.55 | **69.13±2.93** | **71.76±1.92** | 71.68±2.03 | 68.05±4.42 | 69.47±3.17 |
| NAS-Bench-201 (ImageNet16-120) | 40.59±5.05 | **41.22±4.56** | 40.69±4.09 | **41.24±3.33** | 40.91±5.45 | **41.64±4.76** |
| NAS-Bench-201 (NinaPro) | 90.13±1.30 | **90.40±1.40** | 90.34±1.83 | **90.76±1.89** | 90.77±1.35 | **90.99±1.38** |
| TransNAS-Bench-101 (Jigsaw) | 90.72±4.55 | **91.88±4.76** | 87.28±15.74 | **89.11±11.73** | 90.97±4.63 | **__92.48±3.18__** |
| TransNAS-Bench-101 (Class object) | 47.33±2.47 | **48.26±2.84** | 47.65±2.50 | **48.71±2.69** | 47.91±2.11 | **__48.48±2.46__** |
| TransNAS-Bench-101 (Class scene) | 59.94±5.30 | **60.52±5.14** | 60.87±5.09 | **61.70±4.11** | 60.90±3.53 | **__61.58±1.73__** |
| **N= 100** | | | | | | |
| NAS-Bench-101 (CIFAR-10) | 91.71±1.32 | **92.06±1.15** | 91.98±2.08 | **91.98±2.01** | 89.67±2.25 | **90.06±1.98** |
| NAS-Bench-201 (CIFAR-10) | 91.10±1.61 | **91.73±1.43** | 92.11±0.98 | **92.52±1.10** | 91.09±1.84 | **91.75±2.38** |
| NAS-Bench-201 (CIFAR-100) | 68.43±2.82 | **69.05±2.64** | 70.38±1.77 | **70.75±1.79** | 69.39±4.50 | **70.01±4.36** |
| NAS-Bench-201 (ImageNet16-120) | 40.34±4.66 | **40.96±4.98** | 42.54±3.93 | **43.34±2.47** | 41.04±4.49 | **41.50±4.42** |
| NAS-Bench-201 (NinaPro) | 90.00±1.91 | **90.22±1.86** | 89.18±1.72 | **__89.53±1.55__** | 91.29±1.00 | **91.36±1.05** |
| TransNAS-Bench-101 (Jigsaw) | 89.95±5.57 | **90.59±4.10** | **__90.58±3.55__** | 85.33±2.21 | **92.00±2.78** | 91.99±2.96 |
| TransNAS-Bench-101 (Class object) | 47.04±4.06 | **__49.46±4.56__** | 48.25±3.11 | **48.87±2.96** | 49.48±0.97 | **49.81±1.77** |
| TransNAS-Bench-101 (Class scene) | 60.94±3.23 | **61.12±4.14** | 61.19±3.10 | **61.50±4.32** | **61.56±0.56** | 61.29±2.85 |
| **N= 1000** | | | | | | |
| NAS-Bench-101 (CIFAR-10) | 91.47±1.36 | **91.57±1.31** | 92.12±2.45 | **92.60±1.73** | 90.54±2.35 | **90.92±2.40** |
| NAS-Bench-201 (CIFAR-10) | 91.25±1.46 | **92.08±1.64** | 92.04±1.22 | **__92.69±1.08__** | 91.00±2.07 | **91.88±2.16** |
| NAS-Bench-201 (CIFAR-100) | 69.32±3.17 | **__70.63±2.68__** | 70.78±1.66 | **__71.52±1.14__** | 69.83±2.62 | **70.40±2.58** |
| NAS-Bench-201 (ImageNet16-120) | 41.42±4.56 | **42.17±4.11** | 42.96±2.78 | **43.79±2.92** | 41.59±3.41 | **42.24±2.03** |
| NAS-Bench-201 (NinaPro) | 90.59±1.73 | **90.61±2.19** | **90.47±1.25** | 90.21±1.34 | 91.49±0.65 | **91.57±0.89** |
| TransNAS-Bench-101 (Jigsaw) | 88.63±4.32 | **90.81±3.93** | **90.60±3.00** | 88.52±11.35 | 92.40±0.98 | **92.57±1.51** |
| TransNAS-Bench-101 (Class object) | 44.64±3.26 | **__45.46±4.21__** | 44.99±3.31 | **45.82±5.97** | 44.98±1.15 | **__45.36±0.89__** |
| TransNAS-Bench-101 (Class scene) | 61.42±3.37 | **61.50±3.51** | 61.47±3.09 | **61.76±4.32** | 61.74±0.43 | **__62.29±0.45__** |

Table 2: Evolutionary search results. Results with averaging (Avg) and statistical (Stat) evaluation are provided. Mann-Whitney U-test is used for the statistical evaluation. Each ranking function is evaluated 10 times for the architecture. For each evaluation pair, the best score is highlighted in **bold**; if the higher score is statistically significant with p-value below 0.05, it is highlighted in **bold underlined**.

| Search Space | Search Alg. | Eigenvalue score | | ReLU dist | | Ensemble | |
|---|---|---|---|---|---|---|---|
| | | Avg | Stat | Avg | Stat | Avg | Stat |
| NAS-Bench-101 (CIFAR-10) | Greedy | 92.76±0.90 | **92.86±0.66** | 92.36±0.31 | **92.54±0.59** | 92.54±1.01 | **92.82±0.77** |
| | REA | 93.02±0.32 | **93.11±0.30** | 93.15±0.15 | **__93.31±0.09__** | 92.99±0.42 | **93.28±0.25** |
| | FreeREA | 93.23±0.47 | **__93.51±0.35__** | 93.29±0.58 | **93.46±0.33** | 93.32±0.53 | **93.52±0.38** |
| NAS-Bench-201 (CIFAR-100) | Greedy | 69.54±0.13 | **__70.01±0.42__** | 70.00±0.03 | **70.06±0.02** | 71.03±0.46 | **71.12±0.58** |
| | REA | 71.64±0.80 | **72.14±0.69** | 71.83±0.61 | **72.25±0.44** | 72.06±0.64 | **72.40±0.75** |
| | FreeREA | 71.98±0.26 | **__73.13±0.07__** | 71.63±1.35 | **72.43±1.04** | 71.89±0.97 | **72.15±0.76** |
| NAS-Bench-201 (ImageNet16-120) | Greedy | 42.68±1.06 | **__44.08±0.57__** | 44.78±1.34 | **45.90±0.71** | 44.11±1.79 | **45.21±0.80** |
| | REA | 44.90±0.76 | **45.18±0.56** | 43.82±1.53 | **44.45±1.06** | 44.23±0.84 | **44.76±0.94** |
| | FreeREA | 44.84±1.17 | **45.52±0.70** | 43.82±1.23 | **44.38±0.92** | 44.26±1.48 | **__45.22±0.74__** |
| TransNAS-Bench-101 (Jigsaw) | Greedy | 92.49±1.11 | **93.68±0.80** | 92.23±0.94 | **92.98±0.69** | 93.06±0.85 | **__94.29±0.41__** |
| | REA | 91.88±0.31 | **93.47±0.23** | 92.12±0.39 | **93.29±0.29** | 93.80±0.41 | **__95.06±0.60__** |
| | FreeREA | 91.79±0.18 | **93.26±0.11** | 92.03±0.17 | **92.97±0.18** | 93.42±0.58 | **__94.99±0.58__** |
| TransNAS-Bench-101 (Class object) | Greedy | 43.89±1.73 | **45.16±1.44** | **46.04±0.07** | 46.01±0.04 | 46.29±3.34 | **__47.19±0.12__** |
| | REA | 46.10±0.09 | **__46.44±0.07__** | 45.99±0.09 | **46.86±0.14** | 46.26±0.08 | **__46.52±0.20__** |
| | FreeREA | 46.73±0.09 | **46.81±0.08** | **46.97±0.06** | 46.97±0.07 | 47.09±0.11 | **47.14±0.09** |

Table 3: Experiemental results on SWAPNAS, MeCo, SynFlow and LogSynFlow for evolutionary search. Results with averaging (Avg) and statistical (Stat) evaluation are provided. Mann-Whitney U-test is used for the statistical evaluation. Each ranking function is evaluated 10 times for the architecture. For each evaluation pair, the best score is highlighted in **bold**; if the higher score is statistically significant with p-value below 0.05, it is highlighted in **bold underlined**.

| Ranking function | Method | NAS-Bench-101 (CIFAR-10) | | NAS-Bench-201 (ImageNet16-120) | |
|---|---|---|---|---|---|
| | | Avg | Stat | Avg | Stat |
| SWAPNAS | REA | 93.05±0.12 | **93.37±0.11** | 44.39±0.98 | **45.74±1.22** |
| | FreeREA | 93.42±0.36 | **93.71±0.32** | 44.75±1.15 | **__45.89±1.01__** |
| MeCo | REA | 93.13±0.54 | **93.52±0.44** | 42.47±0.84 | **42.81±0.81** |
| | FreeREA | 93.21±0.45 | **93.51±0.39** | 42.88±0.79 | **43.23±0.85** |
| SynFlow | REA | **93.69±0.38** | 93.56±0.31 | 44.29±1.02 | **44.57±0.99** |
| | FreeREA | **93.80±0.24** | 93.74±0.32 | **44.98±0.89** | 44.74±1.21 |
| LogSynFlow | REA | **93.76±0.36** | 93.62±0.28 | **45.03±0.82** | 44.91±1.04 |
| | FreeREA | 93.76±0.25 | **93.77±0.36** | 45.06±0.90 | **45.16±0.95** |

ranking function usage. A ranking function is evaluated ten times on every queried architecture. In order to reduce the number of experiments we do not test combinations of ranking function. We provide results for averaged rank value and for the proposed statistical ranking approach. The 5% significance level is used for rejecting the null hypothesis when conducting the Mann-Whitney U-test. We repeat each experiment 100 times, computing the mean value of the accuracy for the selected architecture and its variation. Our findings demonstrate that our statistical approach is a better way to use the multiple outcomes than simple averaging. As we can see in Table 1, the proposed statistical method leads to improved performance for almost every ranking function and search space. The average accuracy improvement is 0.49. The improvement is small, but there is a negligible additional computational overhead. Our experiments do not expose a significant difference between the improvement for low- versus high-variance ranking functions. We observed that the largest search improvement is achieved on the NAS-Bench-201 search spaces, but that could be due to the fact that those search spaces are small. Less improvement is observed for the largest search space NAS-Bench-101 (CIFAR-10).

## 5.3 Evolutionary search

We also compare the performance of the standard averaging approach with that of the proposed statistical evaluation when each is used in conjunction with evolutionary search. As before, the 5% significance level is used for rejecting the null hypothesis of the Mann-Whitney U-test. We apply caching for both approaches. In the case of averaging, we cache the average of the ranking function output over 10 evaluations. For the statistical evaluation we cache the individual outcomes of the 10 evaluations. Whenever two architectures are statistically compared, we store the outcome, so that the procedure does not need to be repeated.

Table 2 presents the evolutionary search results on the NAS-Bench and TransNAS-Bench search spaces. We demonstrate results on Eigenvalue score and ReLU Hamming distance. We also include an ensemble of those that is formulated as a sum with components individually MinMax normalized over the observed architecture ranks. We observe that statistical evaluation consistently improves the search quality. Although the particular improvement depends on the combination of the search space and ranking function, we did not observe a case where applying a statistical evaluation leads to a poorer search result. We do not observe a significant difference between the improvement for low- versus high-variance ranking functions. In our experiments with evolutionary search, Eigenvalue score (low variance) has the average improvement of 0.73, whereas for RELU dist (high variance) the average improvement is 0.60. We observed that FreeREA demonstrates the best results in most cases, thus emphasising the importance of crossover in evolutionary search. However, in the averaging case the difference between FreeREA and REA is typically less than one standard deviation. This may make REA more suitable for some practical applications as it requires only to define a mutation operation. FreeREA experiences the largest performance increase with statistical

evaluation, and clearly outperforms REA. In some cases, we observed that the greedy algorithm outperforms the others. It is not an expected outcome, but we assume it is due to randomness of the experimental setup. Similar to other zero-shot works (Cavagnero et al., 2023a; Wang et al., 2024), our experiments demonstrate that evolutionary search is more stable and efficient than random search. The only search space where random and evolutionary search exhibit similar performance is TransNAS-Bench-101, and we attribute this to the extremely small size of this search space (see Appendix A).

In Table 3 we demonstrate that our approach is generalisable to more recently proposed ranking functions. SWAPNAS (Peng et al., 2024) and MeCo (Jiang et al., 2024) are Zero-Shot ranking functions that aim to achieve the highest correlation with the accuracy of the architecture in the NAS-Bench search spaces. Table 3 shows that the proposed methodology can easily be adapted to the new ranking functions. In addition, we provide results on data-agnostic SynFlow (Abdelfattah et al., 2021) and LogSynFlow (Cavagnero et al., 2023a). Unlike any other ranking functions presented in this paper, the variation of SynFlow and LogSynFlow is induced only by random initialisation of the evaluated architecture. The data-agnostic approaches present a challenge for our statistical evaluation. Averaging demonstrates performance comparable or higher to that of our statistical evaluation on them. More results for SWAPNAS, MeCo, SynFlow and LogSynFlow are demonstrated in the Appendix C.

## 6 Conclusion and Future work

In this work, we proposed a statistical approach for comparing architectures during a NAS zero-shot search procedure. Through experiments using both random and evolutionary search, we provided evidence that the proposed approach leads to better search results than simply averaging the outcomes of a batch of evaluations. We also investigated the variation of a wide selection of zero-shot ranking functions for several search spaces and observed that those which exhibit less variability tend to produce better and more stable search outcomes. Given these observations, we can recommend that zero-shot search is conducted using statistical architecture comparison.

Every ranking function evaluation for a given architecture can be considered equivalent to sampling from the distribution. In our work, we handled such sampling with statistical testing in order to retain compatibility with the existing pipelines and search agorithms. Formulating a zero-shot neural architecture search as a stochastic optimisation problem that interprets the output of a ranking function not as a score, but as a distribution could be a potential research direction.

We have observed that the statistical method does not consistently benefitial for data-agnostic SynFlow and LogSynFlow, while their performance is competitive with other ranking functions discussed in the paper. This suggests that an alternative strategy to the statistical approach is to develop techniques that are minimally affected by the data batch and initialisation.

An additional important source of performance variability that we have not discussed is the search space. Our experiments show that the suitability of a ranking function is highly dependent on the search space. For one search space, a particular ranking function might be the best choice; on a different search space, it might perform poorly. Due to this variability, we cannot predict how our conclusions will generalise to a new search space.

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

## A    NAS search space benchmarks

**NAS-Bench-101** Ying et al. (2019) includes the performance and training statistic of the 423k architectures on CIFAR-10. It has three cells. Each cell containing 7 nodes each with a pool of operations consists of: "3x3 conv", "1x1 conv" and "3x3 max-pooling".

**NAS-Bench-201** (Dong & Yang, 2019) has 15625 architectures in the search space and provides performance for CIFAR-10, CIFAR-100, and ImageNet-16-120. It has 5 cells. Every cell has 4 nodes. The list of operations includes "zeroize", which is equivalent to cutting the connection, "skip connection", "3x3 conv", "1x1 conv" and "3x3 avg-pooling". This is the most popular benchmark as it is relatively small and contains measurements for several datasets. **NAS-Bench-360** (Tu et al., 2022) extends this benchmark by additional datasets. In this work, we are using the NinaPro DB5 dataset from NAS-Bench-360. For consistency, we refer to it as NAS-Bench-201 (NinaPro).

**TransNAS-Bench-101** (Duan et al., 2021) is a tabular benchmark that consists of two search spaces evaluated on seven tasks. The architecture of the neural network in the benchmark is a composition of a searchable encoder and a decoder tailored to specific tasks. The macro search space suggests searching for the number of blocks, the size for downsampling, and the count of output channels per block. The micro search space is composed of four nodes, similarly to NAS-Bench-201. Four operations are used: "zeroize", "skip connection", "3x3 conv block" and "1x1 conv block", where every convolution block is a sequence of ReLU, convolution, and batch normalisation. In our project, we are only using the micro search space that consists of 4096 graphs.

## B    Evolutionary search algorithms

Here we present pseudocode for all evolutionary algorithms used in this project. Since the greedy evolutionary algorithm and FreeREA are primarily based on REA, we provide full pseudocode for REA (see Algorithm 2) and parts that are different for FreeREA (see Algorithm 4) and greedy evolutionary search (see Algorithm 3). The core notation for all the algorithms is chosen to be close to REA notation, as presented by Real et al. (2019). For the standard implementation, the highest ranked candidate and the top-2 are computed by maximizing over the averaged ranking function output. For our implementation, we use statistical comparison with stat-max and stat-top-2 from Algorithm 1.

We designed a greedy evolutionary search algorithm in order to further demonstrate how ranking function variance influences evolutionary search algorithm performance. The primary difference between greedy evolutionary search and REA lies in the mutation operation. Instead of randomly mutating a parent candidate as in REA, it greedily explores a list of all possible mutations. Therefore, it is expected to experience a greater impact from the variance introduced by a ranking function.

In all our experiments, we are utilising an evaluation budget ($C$ at line 9 of Algorithm 2) that defines how many architectures can be evaluated in total. As one can see, the three evolutionary algorithms are spending the budget at different rates. REA evaluates a single candidate per iteration. FreeREA evaluates three and greedy search evaluates all the mutations (i.e., the neighbourhood on the search hypergraph).

## C    Additional experiments with ranking functions

In this section, we present an evaluation with SWAPNAS (Peng et al., 2024), MeCo (Jiang et al., 2024), SynFlow (Abdelfattah et al., 2021) and LogSynFlow (Cavagnero et al., 2023a). SWAPNAS and MeCo are recent Zero-Shot ranking functions that aim for the highest correlation with the architecture accuracy on the NAS-Bench search spaces. SynFlow and LogSynFlow are data-agnostic ranking functions. Due to

---

**Algorithm 2** Regularised evolutionary algorithm (REA)

---

1: $population \leftarrow \emptyset$
2: $history \leftarrow \emptyset$
3: **while** $|population| < P$ **do** $\qquad\qquad\qquad\qquad\qquad\qquad$ ▷ Initialize population of size $P$
4: $\quad arch \leftarrow$ ARCHSAMLE() $\qquad\qquad\qquad\qquad\qquad\qquad$ ▷ Randomly sample architecture
5: $\quad rank \leftarrow$ EVALARCH($arch$) $\qquad\qquad\qquad\qquad$ ▷ Evaluate ranking function for architecture
6: $\quad history \leftarrow history \cup \{(arch, rank)\}$
7: $\quad population \leftarrow population \cup \{(arch, rank)\}$
8: **end while**
9: **while** $|history| < C$ **do** $\qquad\qquad\qquad\qquad\qquad\qquad\qquad\qquad$ ▷ Evolve for $C$ cycles
10: $\quad random\_candidates \leftarrow \emptyset$
11: $\quad$ **for** $i \leftarrow 1, S$ **do**
12: $\qquad candidate \leftarrow$ RANDOM(population) $\qquad\qquad$ ▷ Sample random element from population
13: $\qquad random\_candidates \leftarrow random\_candidates \cup \{candidate\}$
14: $\quad$ **end for**
15: $\quad parent \leftarrow$ highest rank candidate out of $random\_candidates$
16: $\quad child \leftarrow$ MUTATE($parent$) $\qquad\qquad\qquad\qquad$ ▷ Sample one of all possible mutations
17: $\quad rank \leftarrow$ EVALARCH($child$) $\qquad\qquad\qquad\qquad$ ▷ Evaluate ranking function for the child
18: $\quad history \leftarrow history \cup \{(child, rank)\}$
19: $\quad population \leftarrow population \cup \{(child, rank)\}$
20: $\quad$ remove the oldest member of the population
21: **end while**
22: $best \leftarrow$ highest rank candidate out of $history$
23: **return** $best$

---

---

**Algorithm 3** Greedy evolutionary search algorithm difference with Algorithm 2

---

$\cdots$
15: $parent \leftarrow$ highest rank candidate out of $random\_candidates$
16: $children \leftarrow$ MUTATEALL($parent$) $\qquad\qquad\qquad\qquad$ ▷ Get list of all possible mutations
17: $child \leftarrow$ highest rank candidate out of $children$
18: $rank \leftarrow$ EVALARCH($child$) $\qquad\qquad$ ▷ Evaluate all children and get the highest rank one
19: add all mutations and their respective ranks to the $history$
$\quad$ /* Proceed to line 19 of Algorithm 2 */
$\quad \cdots$

---

---

**Algorithm 4** Free regularised evolutionary algorithm (FreeREA) difference with Algorithm 2

---

$\cdots$
15: $parent_1, parent_2 \leftarrow$ top-2 highest rank candidates out of $random\_candidates$
16: $child_1 \leftarrow$ MUTATE($parent_1$)
17: $rank_1 \leftarrow$ EVALARCH($child_1$)
18: $child_2 \leftarrow$ MUTATE($parent_2$)
19: $rank_2 \leftarrow$ EVALARCH($child_2$)
20: $child_3 \leftarrow$ CROSSOVER($parent_1, parent_2$) $\qquad\qquad$ ▷ Perform crossover between two parents' genes
21: $rank_3 \leftarrow$ EVALARCH($child_3$)
22: $history \leftarrow history \cup \{(child_1, rank_1), (child_2, rank_2), (child_3, rank_3)\}$
23: $population \leftarrow population \cup \{(child_1, rank_1), (child_2, rank_2), (child_3, rank_3)\}$
24: remove top-3 the oldest member of the population
$\quad$ /* Proceed to line 21 of Algorithm 2 */
$\quad \cdots$

---

Table 4: Evolutionary search on SWAPNAS and MeCo. Results with averaging (Avg) and statistical (Stat) evaluation are provided. Mann-Whitney U-test is used for the statistical evaluation. Each ranking function is evaluated 10 times for the architecture. For each evaluation pair, the best score is highlighted in **bold**; if the higher score is statistically significant with p-value below 0.05, it is highlighted in **bold underlined**.

| Search Space | Search Alg. | SWAPNAS | | MeCo | |
|---|---|---|---|---|---|
| | | Avg | Stat | Avg | Stat |
| NAS-Bench-101 (CIFAR-10) | REA | 93.05±0.12 | **93.37±0.11** | 93.13±0.54 | **93.52±0.44** |
| | FreeREA | 93.42±0.36 | **93.71±0.32** | 93.21±0.45 | **93.51±0.39** |
| NAS-Bench-201 (CIFAR-100) | REA | 71.77±0.49 | **72.54±0.56** | 71.31±1.37 | **71.54±1.11** |
| | FreeREA | 71.70±1.02 | **72.81±0.98** | **71.86±0.87** | 71.86±0.99 |
| NAS-Bench-201 (ImageNet16-120) | REA | 44.39±0.98 | **45.74±1.22** | 42.47±0.84 | **42.81±0.81** |
| | FreeREA | 44.75±1.15 | **45.89±1.01** | 42.88±0.79 | **43.23±0.85** |
| TransNAS-Bench-101 (Jigsaw) | REA | 92.36±0.44 | **93.17±0.30** | 92.43±0.14 | **93.64±0.13** |
| | FreeREA | 92.29±0.18 | **93.04±0.21** | 91.68±0.20 | **93.55±0.09** |
| TransNAS-Bench-101 (Class object) | REA | 45.96±0.11 | **46.80±0.12** | 46.73±0.09 | **46.99±0.08** |
| | FreeREA | 47.01±0.07 | **47.09±0.05** | 46.60±0.10 | **47.11±0.12** |

Table 5: Evolutionary search on SynFlow and LogSynFlow. Results with averaging (Avg) and statistical (Stat) evaluation are provided. Mann-Whitney U-test is used for the statistical evaluation. Each ranking function is evaluated 10 times for the architecture. For each evaluation pair, the best score is highlighted in **bold**; if the higher score is statistically significant with p-value below 0.05, it is highlighted in **bold underlined**.

| Search Space | Search Alg. | SynFlow | | LogSynFlow | |
|---|---|---|---|---|---|
| | | Avg | Stat | Avg | Stat |
| NAS-Bench-101 (CIFAR-10) | REA | **93.69±0.38** | 93.56±0.31 | **93.76±0.36** | 93.62±0.28 |
| | FreeREA | **93.80±0.24** | 93.74±0.32 | 93.76±0.25 | **93.77±0.36** |
| NAS-Bench-201 (CIFAR-100) | REA | **71.52±0.50** | 71.34±0.47 | **71.94±0.50** | 71.53±0.62 |
| | FreeREA | **71.62±0.73** | 72.46±1.04 | **72.23±0.68** | 73.14±0.39 |
| NAS-Bench-201 (ImageNet16-120) | REA | 44.29±1.02 | **44.57±0.99** | **45.03±0.82** | 44.91±1.04 |
| | FreeREA | **44.98±0.89** | 44.74±1.21 | 45.06±0.90 | **45.16±0.95** |
| TransNAS-Bench-101 (Jigsaw) | REA | 91.84±0.17 | **92.71±0.41** | 91.92±0.33 | **92.49±0.28** |
| | FreeREA | 91.46±0.25 | **92.22±0.53** | 91.46±0.20 | **91.89±0.62** |
| TransNAS-Bench-101 (Class object) | REA | 46.51±0.03 | **46.82±0.10** | 46.50±0.06 | **46.82±0.10** |
| | FreeREA | 46.75±0.07 | **47.12±0.05** | 46.66±0.04 | **47.03±0.09** |

Table 6: Random search on $N$ sampled architectures; coefficient of variation (CV) is used as a ranking function. Statistical evaluation of the ranking function (Stat) is provided for the reference. Each ranking function is evaluated 10 times for the architecture on a batch of 64. We provide mean values and standard deviations over 100 runs. Each ranking function is evaluated 10 times for the architecture. For each pair of results, the best score is highlighted in **bold**; if the higher score is statistically significant with p-value below 0.05, it is highlighted in **bold underlined**.

| Search Space | Eigenvalue score | | ReLU dist | | NTK(cond) | |
|---|---|---|---|---|---|---|
| | CV | Stat | CV | Stat | CV | Stat |
| **N= 10** | | | | | | |
| NAS-Bench-101 (CIFAR-10) | 91.30±3.04 | **91.71±2.22** | **91.74±1.74** | 91.63±1.51 | 91.15±1.68 | **91.24±1.63** |
| NAS-Bench-201 (ImageNet16-120) | 39.58±6.11 | **41.22±4.56** | 37.02±5.80 | **41.24±3.33** | **42.59±4.73** | 41.64±4.76 |
| TransNAS-Bench-101 (Class object) | 47.24±5.00 | **48.26±2.84** | **48.72±2.32** | 48.71±2.69 | 46.72±5.44 | **48.48±2.46** |
| **N= 100** | | | | | | |
| NAS-Bench-101 (CIFAR-10) | 91.36±1.56 | **92.06±1.15** | 91.61±2.68 | **91.98±2.01** | **90.18±1.83** | 90.06±1.98 |
| NAS-Bench-201 (ImageNet16-120) | 35.98±5.59 | **40.96±4.98** | 35.97±5.92 | **43.34±2.47** | 38.21±5.29 | **41.50±4.42** |
| TransNAS-Bench-101 (Class object) | 45.27±6.20 | **49.46±4.56** | 45.30±6.37 | **48.87±2.96** | 48.24±2.77 | **49.81±1.77** |
| **N= 1000** | | | | | | |
| NAS-Bench-101 (CIFAR-10) | 91.21±1.51 | **91.57±1.31** | **92.64±1.51** | 92.60±1.73 | 89.04±1.89 | **92.92±2.40** |
| NAS-Bench-201 (ImageNet16-120) | 35.26±5.50 | **42.17±4.11** | 35.51±5.35 | **43.79±2.92** | 35.20±5.69 | **42.24±2.03** |
| TransNAS-Bench-101 (Class object) | 44.79±6.72 | **46.46±4.21** | 47.39±3.83 | **47.82±5.97** | 45.20±6.40 | **49.36±0.89** |

constant input, SynFlow and LogSynFlow have no randomness induced by the input, therefore, resulting in less overall variation. Note, however, that there is still variation due to the initialization of the architecture. We present evolutionary search results for averaging over $V = 10$ evaluations and the batch size $B = 64$ for SWAPNAS and MeCo. We used REA and FreeREA evolutionary search algorithms for these experiments. For statistical evaluation with Algorithm 1 the Mann-Whitney U-test is used with the confidence threshold set to 0.05. In Table 4 we present the means and standard deviations for SWAPNAS and MeCo calculated over 10 runs. These results show that the proposed methodology can easily be adapted to recently published ranking functions that exhibit high correlation with architectural accuracy. The proposed statistical approach outperforms in all but one setting, with statistical significance in several cases. In Table 5 we present the means and standard deviations for SynFlow and LogSynFlow calculated over 10 runs.

We see that the statistical approach is not consistently advantageous for these data-agnostic approaches. This is in part due to the reduction in variability by the removal of dependence on the data batch. A secondary factor is that with the existing (author-provided) implementations of SynFlow and LogSynFlow we observe occasional dramatic outliers due to gradient explosion for some initializations. This could potentially be addressed by an alternative implementation. When these outliers arise, both the statistical and averaging approaches can become equivalent to a random architecture selection, inhibiting the effectiveness of the search. Since SynFlow and LogSynFlow are competitive with other ranking functions that we present in the paper, this suggests that an alternative strategy to the one we present in this paper is to develop techniques that are minimally affected (or even not affected) by the data batch and initialisation.

# D   Coefficient of variations as a ranking function

A naive way to incorporate a coefficient of variation into the search process is to utilise it as a ranking function. Given a ranking function $r(arch, d)$ we construct a coefficient of variation estimator as

$$r_{CV}(arch, \mathcal{D}) = Mean(\mathcal{M}) + CV(\mathcal{M}) \quad \text{for} \quad \mathcal{M} = \{r(arch, d_v)\}_{v=1}^{V}, \tag{2}$$

where $d_v \in \mathcal{D}$ is a batch of data and $CV(\mathcal{X}) = \frac{Var(\mathcal{X})}{Mean(\mathcal{X})}$.

As shown in Fig. 3, the ranking function coefficient of variation exhibits a considerable Kendall-$\tau$ correlation with accuracy. Moreover, in almost all cases this correlation is negative, which makes coefficient of variation a suitable ranking function for many metrics.

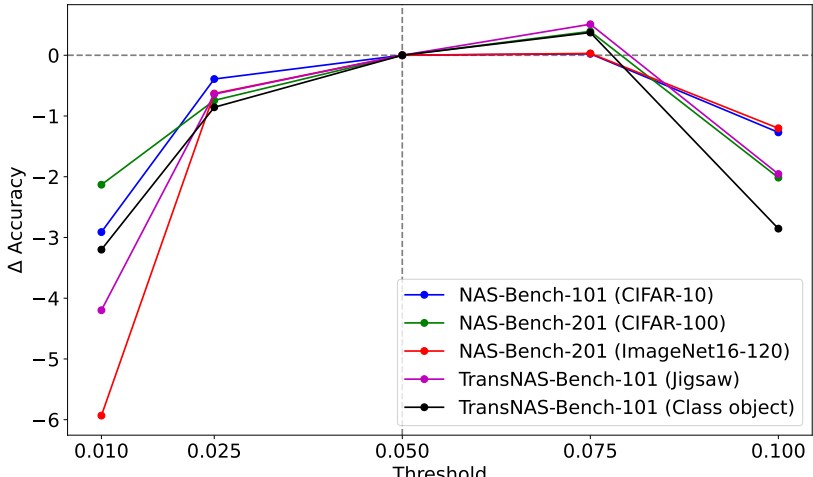

Figure 5: Significance level ablation of the p-value threshold. The Eigenvalue score is used with FreeREA search algorithm. The accuracy is presented with respect to the accuracy of 0.05 threshold (see Table 2).

To explore this further, we conduct an experiment for several search spaces where we use random search and sample $N$ architectures. To stabilize search performance, we use Min-Max normalization individually for the mean term and the CV term when computing $r_{CV}$ in equation 2.

We demonstrate some results in Table 6. Although in some cases on NAS-Bench-101 $r_{CV}$ exhibits performance comparable to statistical evaluation, its performance is consistently worse on other search spaces, in some cases by a considerable margin. In many cases, modification of the ranking function is harmful to performance compared to simply using the mean. We attribute this to the correlation between the coefficient of variation and a ranking function averaged value (see Fig. 4). We have noticed that the Kendall-$\tau$ correlation between the coefficient of variation of a ranking function and its mean value is the highest for the low-variance ranking functions that are the most suitable for zero-shot search (see Fig. 4). For those ranking functions, the coefficient of variation does not provide much additional useful information for the search, but can introduce additional variability, thus making the results inconsistent.

## E  Significance level ablation

In the main body of the article we use the significance level of 0.05 for the Algorithm 1 threshold. This value was selected as the most commonly used for statistical testing. However, this threshold is a hyperparameter that can be optimised. Fig. 5 shows that the optimal threshold lies within the range from 0.025 to 0.075. The lower threshold seems to be excessively stringent, resulting in reduced performance. Conversely, the higher threshold is overly lenient, permitting false cases of stochastic dominance.

## F  Caching in the evolutionary search

We would like to demonstrate the importance of caching. Unlike random search, where all architectures are sampled once, in evolutionary search, the same architecture can be encountered multiple times due to mutation or crossover. Due to the variations in the ranking function output, reevaluation of a previously encountered architecture can lead to a performance degradation in the search. We demonstrate this in Table 7. We report mean values and standard deviations for accuracies over 10 runs. A randomly sampled batch of 64 with randomly initialised architecture weights is used to evaluate an architecture. In the case of caching, we evaluate the architecture only once, when it is first encountered, and then save the ranking function output. Subsequently, if the architecture is encountered again, the cached value of the ranking function is used in comparisons. In the 'on-the-fly' alternative, the architecture is reevaluated every time it is encountered. We evaluate an architecture with a ranking function 10 times and average the outputs in

Table 7: Accuracy comparison between cached rank mapping (Cached) and computation on-the-fly (On-the-fly). Variation of ranking function influence on NAS-Bench-101/201. We provide mean values and standard deviations over 10 runs. For each comparison pair, the best score is highlighted in **bold**; if the higher score is statistically significant with p-value below 0.05, it is highlighted in **bold underlined**.

| Search Space | Search Alg. | Eigenvalue score | | | ReLU dist | | |
|---|---|---|---|---|---|---|---|
| | | Cached | Hybrid | On-the-fly | Cached | Hybrid | On-the-fly |
| NAS-Bench-201 (CIFAR-10) | Greedy | **92.76±0.90** | 92.12±1.18 | 92.30±0.90 | **92.36±0.31** | 91.65±4.29 | 91.91±3.85 |
| | REA | **93.02±0.32** | 92.56±2.86 | 91.98±4.11 | **93.15±0.15** | 91.77±1.45 | 90.74±4.24 |
| | FreeREA | **93.23±0.47** | 92.51±1.43 | 92.43±2.09 | **93.29±0.58** | 91.94±4.02 | 91.27±7.50 |
| NAS-Bench-201 (CIFAR-100) | Greedy | **69.54±1.01** | 67.21±1.48 | 65.42±1.37 | **70.00±0.03** | 66.90±1.78 | 65.83±2.11 |
| | REA | **71.54±0.80** | 67.64±1.09 | 67.42±1.37 | **71.83±0.61** | 67.95±1.07 | 68.02±1.50 |
| | FreeREA | **71.98±0.26** | 69.43±1.03 | 69.44±1.10 | **71.63±1.35** | 68.44±1.40 | 68.89±1.42 |
| NAS-Bench-201 (ImageNet16-120) | Greedy | **42.68±1.06** | 39.68±4.40 | 41.14±3.29 | **44.78±1.34** | 41.75±5.62 | 42.54±4.75 |
| | REA | **44.90±0.76** | 43.51±4.22 | 43.55±4.71 | **43.82±1.53** | 42.12±2.48 | 41.34±7.19 |
| | FreeREA | **44.84±3.17** | 42.33±3.52 | 40.62±5.28 | **43.82±1.23** | 42.71±2.70 | 42.58±5.24 |

both cases. In addition, we introduce a 'hybrid' method where evaluations are accumulated. The first time we encounter an architecture, we add 10 evaluations. Every subsequent time an architecture is encountered, three new evaluations are added. Thus, our setup is similar to the most typical case of an evolutionary search. To consistently compare evolutionary algorithms, we use an evaluation budget and cap it at 1000 evolution cycles. We would like to highlight that we limit the number of evolution cycles and not the number of architecture evaluations (see line 21 of Algorithm 2). In this setting, REA has no exploration advantage over Greedy search, which evaluates significantly more architectures per cycle. Both methods explore a similar number of architectures, because the search process of REA is more effective.

In addition to REA and FreeREA evolutionary searches, we also report results for a greedy evolutionary search. It is similar to REA, but evaluates all possible mutations of the sampled exemplar architecture and selects the best according to the ranking function value. We specially designed it to exclude the randomness introduced by the mutation process. Greedy evolutionary search is only used for benchmarking purposes and is impractical in a real-world setting. See Appendix B for details.

We observe that caching, which is typically applied primarily with the motivation of reducing computational overhead, always improves the search performance. Moreover, caching makes the results more stable in the majority of the cases, i.e., reduces the standard deviation. We conjecture that this performance improvement arises because the use of caching means that previous search decisions are not overturned. In the case of 'on-the-fly' evaluation, the search can initially decide that an architecture is inferior, but then reverse that decision when the architecture is encountered a second time. That can be clearly seen for a greedy evolutionary search, which is designed to maximise the number of re-evaluations. The 'hybrid' approach improves the stability, but results in poorer performance compared to the 'cached' approach. The efficiency of the search is improved by committing to a decision. The observed performance improvement is less for low-variance ranking functions, as their output is more consistent, leading to fewer changed decisions. Table 7 also shows how the selection of low variance ranking functions can significantly improve the quality of search results.

## G    Discussion on meaningful zero for ranking functions

In our work, a coefficient of variation (see equation 1) was used to compare the variability of the commonly used ranking functions (see Fig. 1). The coefficient of variation requires that each ranking function has a meaningful zero. In our work, we use the following ranking functions: Frobenius norm of NTK, mean value of NTK, Label-Gradient Alignment, Eigenvalue score, ReLU Hamming distance, condition number of NTK, EPE-NAS, Fisher, Snip, MeCo, SynFlow, and LogSynFlow.

We have assessed the concept of a meaningful zero for each ranking function listed above. Here are our findings.

- The Frobenius norm of NTK is always bigger than zero, and since it is a norm, it clearly has a meaningful zero.

- The mean value of NTK is a mean of all elements for a positive semidefinite matrix. Since the NTK is positive-semidefinite (Jacot et al., 2018), the mean value is always non-negative, implying that there is a meaningful zero.

- Label-Gradient Alignment is computed from the NTK $H \in \mathbb{R}^{B \times B}$ and label-alignment matrix $L \in \{0,1\}^{B \times B}$ (see (Mok et al., 2022) for details). Since the NTK is positive-semidefinite, LGA is bounded from below. The metric has a meaningful zero.

- The Eigenvalue score is computed using the eigenvalues of NTK $H$, which are all non-negative. The Eigenvalue score has a meaningful zero (corresponding to the extreme case of a zero-valued kernel).

- The ReLU Hamming distance is based on the comparison of ReLU activation patterns for different inputs. Since it is evaluated based on distances, it has a meaningful zero corresponding to the case when the activations match exactly for all inputs.

- EPE-NAS score is the class-wise sum of absolute values, which is non-negative. It has a meaningful zero.

- The condition number of NTK is always positive and greater than or equal to one. Although the zero value cannot be achieved, it is a meaningful value in the sense that the zero point is not arbitrary. We can shift or scale the condition number and retain its fundamental meaning.

- Fisher is a layer-wise sum over squares. It is non-negative and has a meaningful zero.

- Snip is a layer-wise sum over absolute values that is non-negative and has a meaningful zero.

- The implementations of both SynFlow and LogSynFlow use the layer-wise summation of absolute values. These values are non-negative, and they both have a meaningful zero.

- MeCo is a layerwise sum of the smallest eigenvalues of Pearson correlation matrices for feature maps; as the matrices are positive-semidefinite by design (see (Jiang et al., 2024) for details), MeCo is non-negative and has a meaningful zero.

## H  Assets

In this work, we used the following software:

- *automl/NASLib* (Mehta et al., 2022) available on GitHub under Apache 2.0 Licence

- *PyTorch* (Paszke et al., 2019) available via PyPI under a custom BSD licence

- *NumPy* (Harris et al., 2020) available via PyPI under a custom BSD licence

- *SciPy* (Virtanen et al., 2020) available via PyPI under a custom BSD licence

The following dataset were used:

- *CIFAR-10/100* (Krizhevsky, 2009) under CC BY 4.0 Licence

- *ImageNet-16-120* (Chrabaszcz et al., 2017) under CC BY 4.0 Licence

- *NinaPro* (Atzori et al., 2012) under CC BY-ND Licence

- Five datasets from *Taskonomy collection* (Zamir et al., 2018) under CC BY 4.0 Licence

