# OpenReview forum: "Variation Matters: from Mitigating to Embracing Zero-Shot NAS Ranking Function Variation"
_TMLR — Accepted by TMLR_

### Review · Reviewer_pja4 · 2024-12-12

**Summary Of Contributions:**

This paper proposes a new method for improving zero-shot Neural Architecture Search (NAS) by explicitly considering the variation in ranking function outputs. As fast ranking functions are inherently noisy, the authors propose to use statistical comparisons for the distribution of ranking function scores across batches, rather than simple comparing menas as done previously. Algorithms to incorporate this statistical comparisons within random and evolutionary searches are included. Experimental results across ranking functions, architectural search spaces, and hyperparameter values support the utility the proposed methods.

**Audience:**

Yes

**Broader Impact Concerns:**

No broader impacts statement is included, but I don't think it is necessary in this work.

**Claims And Evidence:**

No

**Requested Changes:**

I have listed specific changes following, with rough labels of Clarity, Evidence, Consistency, and Typos. In the paper's current state, I do not recommend acceptance. If all of the following changes are sufficiently addressed, I will change my answer for Claims and Evidence and recommend this work for acceptance.

[Clarity] Abstract: "In the zero-shot version, we use..."- Because you later use "we" to refer to what is done in this paper, avoid using it here to help distinguish that you referring to the general community of zero-shot NAS. For example, "In the zero-shot version, fast ranking functions enable the comparison of architectures without training them."

[Clarity] Section 2.1, paragraph 1: "Random search is the fastest search algorithm"- "Fastest" could imply "least compute to find an acceptable solution". I think you rather mean that the algorithm itself has the least amount of overhead or is the simplest.

[Evidence] Section 2.1, paragraph 1: "Random search suffers the most from variability in the ranking function" - Why? Do you have evidence, whether explanation and/or citations, to support this?

[Evidence] Section 2.1, paragraph 3: "This type of search ... in the ranking function." Again, why? Do you have evidence, whether explanation and/or citations, to support this?

[Clarity] Section 2.1, paragraph 3: "...requires that the architectural search space is specified as a hyper-graph" - Why is this a problem? Are some architectural search spaces not specifiable as a hyper-graph?

[Typo] Section 2.2, paragraph 2: "...for a sampled batch of data Xiong et al. (2020)." should be a parenthetical citation.

[Clarity] Section 3, paragraph 1: "A neural architecture search space can be represented as a graph..." - Do you need to define it as a graph here, especially when graph-based search is not done in this work?

[Typo] Section 4.1, paragraph 1: "...a score to an architecture a batch of data..." ->  "...a score to an architecture, a batch of data..."

[Typo] Section 4.1, paragraph 1: "the data set," -> "the dataset,"

[Clarity] Section 4.1, paragraph 1: "with the extent of the variation being dependent on ... and the task" - Some ranking functions may not be affected by the task (beyond the dataset). For example, ReLU distance is not affect for different tasks on the same dataset, such as using MNIST as the input dataset and predicting the digit label or the parity as the task.

[Consistency] Section 4.1, paragraph 2: The coefficient of variation requires that the ranking function has a meaningful zero. Is this true of all ranking functions considered?

[Evidence] Section 4.2, Algorithm 1: You seem to use a greedy sequential application of the Mann-Whitney-U test. Are you sure this is rigorous? Will it always give the same output given different orderings? Could there be other statistical tests designed for more than pairwise comparison?

[Typo] Section 4.2, paragraph 2: Extraneous newline between "...for all k" and "(and ...".

[Consistency] Section 4.2, paragraph 4: Is $M$ different from the $K$ used in Section 4.1? Also, before $N$ seemed to be all architectures in a search space while here it is only sampled architectures?

[Consistency] Section 4.2, paragraph 5: "...removing the lowest ranked exemplar..." In Appendix B, the oldest population member is removed?

[Clarity] Section 4.2, paragraph 6: You should explicitly write that 3 population members are removed when the new 3 are added.

[Clarity, Consistency] Section 5, Table 1-3: These tables are very dense and hard to quickly parse. I would recommend, for each pair (Avg vs Stat or Cached vs. On-the-fly), bolding the side that has a higher mean and underlining it if it is statistically significantly better. It seems like most of your results have a better mean than the baselines, but likely aren't statistically significantly better. This is an important caveat that you should explicitly address, especially sinc eyour proposed methods are statistics-based.

[Clarity] Section 5.1, Figures 1-2: Why is $B$ repeatedly defined and $K$ is not?

[Evidence] Section 5.1, last paragraph: "In experiments (see Appendix C) we observe that the search performance improvement of this naive scheme is indeed considerably greater for these ranking functions." If I understand correctly, this is only true for 1 experiment pair out of 9, and that only 1 of "these" ranking functions have results shown.

[Clarity, Evidence] Section 5.3, paragraph 1: I would recommend moving this section after the results and discussion in the second half of this subsection, since Table 3 is more of a main result than Table 2, which is more of an ablation study. Additionally, this study seems rather contrived: given an evaluation budget, the cached method should intuitively work better than the on-the-fly method. What if you compare them with a model budget instead? Also, what about a hybrid method, where when an architecture is revisited, it is re-evaluated K more times and these are appended to the existing evaluations?

**Strengths And Weaknesses:**

The idea of explicitly addressing ranking function variation through statistical comparison is novel in zero-shot NAS, to my knowledge. The experiments cover a range of ranking functions, tasks, and archtoectural search spaces, covering a breadth of empirical evidence for the proposed method.

However, not all claims are are well-supported with evidence, whether thorough/clear explanation, results, and/or citations. The specific ones that need better evidence are listed in Requested Changes. I have also noted some issues with clarity and consistency.

---

> ### Author Response · Authors · 2025-01-09
> **Reply to Reviewer pja4 Part 1**
>
> We greatly appreciate your thorough assessment of our manuscript and the valuable feedback provided.
> Due to the significant number of requested changes, some minor requests that lead to straightforward changes are not directly responded to in the following text, but have been directly resolved in the revised paper (highlighted in blue), by directly following the helpful suggestions.
>
> ## **Consistency**
> **Meaningful zero**
>
> We have assessed the concept of meaningful zero for each ranking function listed above.
> Here are our findings.
>
> * The Frobenius norm of NTK is always bigger than zero, and since it is a norm, it clearly has a meaningful zero.
> * The mean value of NTK is a mean of all elements for positive-semidefinite matrix, since NTK is positive-semidefinite~[6], which is always nonnegative, and has a meaningful zero.
> * Label-Gradient Alignment is computed from the NTK $H \in \mathbb{R}^{B\times B}$ and label-alignment matrix $L \in \{0,1\}^{B\times B}$ (see [5] for details). Since the NTK is positive-semidefinite, LGA is bounded from below. The metric has a meaningful zero.
> * The Eigenvalue score is computed using the eigenvalues of NTK $H$, which are all non-negative. The Eigenvalue score has a meaningful zero (corresponding to the extreme case of a zero-valued kernel).
> * The ReLU Hamming distance is based on the comparison of ReLU activation patterns for different inputs. Since it is evaluated based on distances, it has a meaningful zero corresponding to the case when the activations match exactly for all inputs.
> * EPE-NAS score is the class-wise sum of absolute values, which is nonnegative. It has a meaningful zero.
> * The condition number of NTK is always positive and greater than or equal to one. Although the zero value cannot be achieved, it is a meaningful value in the sense that the zero point is not arbitrary. We cannot shift or scale the condition number and retain its fundamental meaning.
> * Fisher is a layer-wise sum over squares, which is nonnegative and has a meaningful zero.
> * Snip is a layer-wise sum over absolute values that is non-negative and has a meaningful zero.
> * The implementations of both SynFlow and LogSynFlow utilise the layer-wise summation of absolute values. These values are non-negative, and they both have a meaningful zero (see https://github.com/SamsungLabs/zero-cost-nas/blob/a43bfbc90c1a02a81fb4397a5e096759e93fbe50/foresight/pruners/measures/synflow.py\#L58 and https://github.com/NiccoloCavagnero/FreeREA/blob/9c2a726cd3adf526c43ae52a515a8fc5ea927fe7/metrics/synflow.py\#L68).
> * MeCo is a layerwise sum of the smallest eigenvalues for Pearson correlation matrices for feature maps; as those matrices are non-negative by design, MeCo is non-negative and has a meaningful zero.
>
> **Section 4.2, paragraph 4: Is $M$ different from the $K$ used in Section 4.1?**
>
> No, $M$ and $K$ are both the number of evaluations.
> We have changed the notation in Sections 4.1 and 4.2 to be consistent throughout the paper.
> We set the number of evaluations to be $V$, the number of architectures to be $N_{SS}$ for the search space $SS$ and the evaluation budget for the random search to be $N$.
>
> **Section 4.2, paragraph 5: "...removing the lowest ranked**
>
> Thank you for bringing this to our attention.
> The REA algorithm replaces the oldest member of the population, not the lowest rank.
> In paragraph 5 we added the sentence
> >We evaluate a ranking function on the mutant $V$times and include the mutated architecture in the population, replacing the oldest ranked exemplar.
>
> In paragraph 6 we added
> >Both mutated exemplars and the resultant crossover exemplar are included into the population replacing three oldest exemplars.

---

> > ### Author Response · Authors · 2025-01-09
> > **Reply to Reviewer pja4 Part 2**
> >
> > ## **Clarity**
> > **Section 2.1, paragraph 1: "Random search is the fastest search algorithm"**
> >
> > hank you for the suggested correction.
> > This line was changed to the
> > > Random search is the search algorithm with the least computation overhead.
> >
> > **Section 2.1, paragraph 3: "...requires that the architectural**
> >
> > Mathematically, every search space can be represented as a hypergraph.
> > Thus, we corrected the sentence to read:
> > > This type of search introduces significant computational overhead as it requires to evaluate the entire hypergraph (Chen et al., 2021)). It is the most challenging to implement and, therefore, is rarely adopted (Chen et al., 2021; Rumiantsev & Coates, 2023; Jiang et al., 2024).
> >
> > **Section 4.1, paragraph 1: "with the extent of the variation**
> >
> > You are correct, some ranking functions are not affected by choosing a task on the same dataset and search space.
> > Some others are not affected by the choice of a dataset for the same task and search space (for instance, Synflow).
> > We have changed the sentence to read:
> > >with the extent of the variation being potentially dependent on ... and the task
> >
> > **These tables are very dense**
> >
> > Thank you for this suggestion.
> > We have revised every table to ensure that they are easier to parse visually.
> > Now, for each pair of results, the higher is highlighted in bold.
> > The statistically significant results are underlined for each pair.
> >
> > ## **Evidence**
> > **Section 2.1, paragraph 1: "Random search suffers the most**
> >
> > To the best of our knowledge, our work is the first to target the variability of the ranking functions.
> > Therefore, no study directly supporting this fact could be cited.
> > The sentence was removed from the paragraph.
> >
> > **Section 2.1, paragraph 3: "This type of search**
> >
> > Prune-based search requires evaluating entire hypergraph for every iteration for every unpruned edge.
> > See the algorithm in (Chen et al., 2021) or (Rumiantsev \& Coates, 2023).
> > It introduced additional overhead compared to the random or evolutionary searches that both only evaluate specific architectures sampled from the hypergraph.
> > Thus, we corrected the sentence to read:
> > >This type of search introduces significant computational overhead as it requires to evaluate the entire hypergraph for every unpruned edge (Chen et al., 2021). It is the most challenging to implement and, therefore, is rarely adopted (Chen et al., 2021; Rumiantsev \& Coates, 2023; Jiang et al., 2024).
> >
> > **Section 4.2, Algorithm 1**
> >
> > Given that we are checking for stochastic dominance, the output of our procedure is not permutation invariant in general.
> > As the first example, let us consider two sets of scores A and B with A stochastically dominating B.
> > The output of Algorithm 1 will be A for both inputs [A,B] and [B,A].
> > Second example, let us consider two sets of scores A and B without stochastic dominance.
> > For input [A, B], the output will be A.
> > For input [B, A], the output will be B.
> > Practically, it is equivalent to a random tie breaker, since the order is decided randomly in our setup.
> >
> > We do not see permutation invariance to be strictly necessary for the search algorithm, which is supported by our experiments.
> > Furthermore, the stochastic variations in the ranking functions inevitably render the selection procedures in all other search algorithms (see Section 2.1) also distinctly susceptible to input permutation.
> > For example, consider the selection step for the evaluation algorithm that uses maximum over averaged scores (see step 15 of Algorithm 2).
> > Assume that two candidates A and B have no stochastic dominance.
> > Given the variation of both candidates' scores, it is a random choice.
> >
> > **Other statistical tests**
> >
> > One of the natural alternatives for the Mann-Whitney U test in our STAT-MAX could be a Dunnett’s test.
> > This test was designed for comparison of multiple measurement sets against a control set, and is still meaningful on as few samples as 10.
> > Using the notation of Algorithm 1, it will predict the p-values of stochastic dominance over the current maximum as $p_i,p_{i+1},...,p_T \leftarrow Dunnet(\text{control}=max,\{r_i,r_{i+1},...,r_T\})$.
> > In order to find the maximum value, this procedure has to be applied iteratively until no stochastic dominance is indicated by any of the p-values.
> > While this test seems more natural for the max function in comparison to our pairwise approach, it is also much more computationally expensive than the pairwise U-test.
> > This is not a problem for a single test that Dunnett’s test was originally designed for; however, in our case hundreds of tests are required, and therefore the computation overhead is important to consider.
> >
> > **Section 5.1, last paragraph**
> >
> > This quote refers to results in Appendix.
> > If you look at Table 5, you will see that in 20 out of 27 cases statistical approach demonstrates higher results than using a ranking function based on CV.

---

> > > ### Author Response · Authors · 2025-01-09
> > > **Reply to Reviewer pja4 Part 3**
> > >
> > > ## **Evidence**
> > > **Section 5.3, paragraph 1 moving**
> > >
> > > Thank you for this suggestion.
> > > We decided to move this section to the appendixes, leaving only Table 3 (now Table 2) in the main body.
> > >
> > > **This study seems rather contrived: given an evaluation budget, the cached method should intuitively work better than the on-the-fly method. What if you compare them with a model budget instead?**
> > >
> > > We agree that given "per-model" cached method intuitively works better since it explores more models during the search procedure.
> > > Please, refer to Algorithm 2 line 9 to see that in our study we introduce not "per-model", but as a number of mutation cycles.
> > > In this case, both methods explore the similar number of architectures.
> > >
> > > **what about a hybrid method, where when an architecture is revisited, it is re-evaluated K more times and these are appended to the existing evaluations?**
> > >
> > > This approach sounds like an interesting extension of an evolutionary search.
> > > We could not identify any paper that implements such a hybrid method.
> > > If possible, we kindly request you to share any of these papers.
> > > Based on our experiments, this hybrid method is expected to worsen the performance similarly to our "on-the-fly" setting.
> > >
> > > ## **References**
> > > 1. Cavagnero, Niccolo, et al. "FreeREA: Training-free evolution-based architecture search." Proceedings of the IEEE/CVF Winter Conference on Applications of Computer Vision. 2023.
> > > 1. Yang, Longxing, et al. "Sweet Gradient matters: Designing consistent and efficient estimator for Zero-shot Architecture Search." Neural Networks 168 (2023): 237-255.
> > > 1. Chen, Wuyang, Xinyu Gong, and Zhangyang Wang. "Neural Architecture Search on ImageNet in Four GPU Hours: A Theoretically Inspired Perspective." International Conference on Learning Representations (ICLR). 2021.
> > > 1.  Rumiantsev, Pavel, and Mark Coates. "Performing Neural Architecture Search Without Gradients." IEEE International Conference on Acoustics, Speech and Signal Processing (ICASSP). 2023.
> > > 1. Mok, Jisoo, et al. "Demystifying the neural tangent kernel from a practical perspective: Can it be trusted for neural architecture search without training?." Conference on Computer Vision and Pattern Recognition (CVPR). 2022.
> > > 1. Jacot, Arthur, et al. "Neural Tangent Kernel: Convergence and generalization in neural networks." Advances in Neural Information Processing Systems (NIPS). 2018.
> > > 1.  Li, Guihong, et al. . "ZiCo: Zero-shot NAS via inverse coefficient of variation on gradients.". International Conference on Learning Representations (ICLR). 2023.

---

> ### Comment · Reviewer_pja4 · 2025-01-15
> **Response to Parts 1-3**
>
> Thank you for thoroughly responding to my review and requests. You did well to consider and comment on all of my concerns in your responses. However, for quite a few "requested changes", you only addressed them in the response and not in the paper. I ask these questions and raise these concerns not for them to only be answered for myself, but to avoid your future audience having the same questions/concerns. Thus, for every "requested change", in addition to your responses posted in these review forums, you should either make a change to the paper or explain why you believe the change should not be made.  For example, in the case of the greedy sequential application of the Mann-Whitney-U test, I recommend that you explain the reasoning given in your response within your paper.
>
> Regarding the "hybrid" method, I was not suggesting that it has already been done, but rather that it could be a third paradigm to compare to Cached and On-the-fly.
>
> New comments:
>
> Abstract: "In the zero-shot version, a fast ranking function are used..." check grammar.
>
> Section 4.2, Algorithm 1: check formatting.

---

> > ### Author Response · Authors · 2025-01-20
> > **Response for additional comments**
> >
> > ## Grammar and Formatting
> > Thank you for bringing our attention to the grammar mistake in the abstract and the formatting issue in Algorithm 1. We have corrected these.
> >
> > ## Regarding the "hybrid" method
> > We will try to execute this experiment and update Appendix F with the new results.
> >
> > ## Address all requested changes in the text
> >
> > We have added a sentence about the meaningful zero to the main text, pointing to a new Appendix G, where there is a more complete discussion, similar to what we provided in the response.
> > We have added the following sentence to Section 4.1.
> >
> > > To ensure the coefficient of variation is a meaningful metric, we have assesed a “meaningful zero” for the ranking functions. We discuss this in detail in Appendix G.
> >
> > The following two paragraphs were added to Section 4.2 to highlight that Algorithm 1 is not permutation invariant and to note the alternative designs.
> >
> > > We summarise this approach as a statistical maximum and statistical top-k (see Algorithm 1).
> > Those procedures iteratively search for a stochastically dominating maximum over the given sets of ranking functions outputs.
> > Given that we are checking for the stochastically dominating maximum, the output of our procedure is not permutation invariant in general.
> > As the first example, let us consider two sets of scores, A and B, with A stochastically dominating B.
> > The output of Algorithm 1 will be A for both inputs [A,B] and [B,A].
> > As a second example, let us consider two sets of scores, A and B, where neither stochastic dominates the other.
> > For input [A, B], the output will be A.
> > For input [B, A], the output will be B.
> > Practically, this is equivalent to a random tie breaker. As random tie breakers are common in search functions, we do not see permutation invariance to be strictly necessary.
> >
> > > We would like to point out that pairwise search is not the only viable design option for the stochastic maximum search.
> > For instance, one of the natural alternatives for the Mann-Whitney-U test could be Dunnett’s test.
> > This test is designed for comparison of multiple measurement sets against a control set.
> > Using the notation of Algorithm 1, it predicts the p-values of stochastic dominance over the current maximum as $p_i,p_{i+1},...,p_T \leftarrow Dunnet(\text{control}=max,\{r_i,r_{i+1},...,r_T\})$.
> > In order to find the maximum value, this procedure has to be applied iteratively until no stochastic dominance is indicated by any of the p-values.
> > However, this design is much more computationally expensive than pairwise comparison.
> > Thus, we use pairwise search to minimise computational overhead.
> >
> > In Section 5.1, last paragraph, we have updated the sentence: "In experiments..." with the following sentence to accommodate your request for clarification.
> >
> > > In experiments (see Table 6 in Appendix D) we observe that the search performance improvement of our
> > approach over this naive scheme is indeed considerably greater for these ranking functions.
> >
> > To clarify the budget setting for the caching experiment, we have added the following sentences to Appendix F.
> >
> > > We would like to highlight that we limit the number of evolution cycles and not the number of architecture evaluations (see line 21 of Algorithm 2).
> > In this setting, REA has no exploration advantage over Greedy search, which evaluates significantly more architectures per cycle.
> > Both methods explore a similar number of architectures, because the search process of REA is more effective.
> >
> > We believe that we have now modified the paper to incorporate all the requested clarifications that were previously in the response.
> > If any comment was overlooked, please inform us and we will be pleased to make suitable modifications. We thank you for the detailed comments which have significantly improved the paper.

---

### Review · Reviewer_6fJQ · 2024-12-14

**Summary Of Contributions:**

This work focuses on analysing the variance in the zero-cost proxy in zero-shot NAS (also referred to as ranking function) subject to different network initialisation, and different inputs. They then, use the variation to slightly improve the random and evolutionary search processes. Intuitively, the more robust a proxy score is (or less the variation), the better the architecture found would be. The experiments show that including the information of variance improves over the standard mean calculation method.

**Audience:**

Yes

**Claims And Evidence:**

No

**Requested Changes:**

Please see the Weakness section

**Strengths And Weaknesses:**

**Strengths**
* The paper is clearly-written, easy to follow and well-structured. The idea of stochastic ordering seems intuitive and experiments show that it works consistently (even if the performance gains are marginal)
* The idea is generalizable, meaning that it theoretically can be applied to any proxy or search space, which offers an opportunity to improve the robustness of the search process.
* This work generally, is appealing to researchers in NAS, mainly because it can be used to boost any proxy (theoretically).

**Weaknesses**
* Although the paper considers some well-established proxies, some additional newer proxy comparisons would be beneficial. Especially, it would be interesting to see the behaviour of proxies like ZiCo [1], MeCo [2], and SWAPNAS [3] when the additional stochastic ordering is implemented.
* The comparative analysis (in Table 1) seems to be insufficient as it just considers three proxies, and out of them, two are just marginally better with the proposed ranking function variation inclusion. This, I believe, is not sufficient to prove the general claim that the method really works with all zero cost proxies. Thus, in the current state, the paper claims and evidence don’t match.
* I think the authors should reformulate the claims in the paper to consider a specific search space and specific proxies, where this method works. Stating that “..this enhances the performance of existing zero-shot architecture search algorithms” is too general and is not experimentally evident.

**References**
[1] Li, G., Yang, Y., Bhardwaj, K., & Marculescu, R. (2023). Zico: Zero-shot nas via inverse coefficient of variation on gradients. arXiv preprint arXiv:2301.11300.
[2] Jiang, T., Wang, H., & Bie, R. (2023). MeCo: Zero-Shot NAS with One Data and Single Forward Pass via Minimum Eigenvalue of Correlation. Thirty-Seventh Conference on Neural Information Processing Systems. Retrieved from https://openreview.net/forum?id=KFm2lZiI7n
[3] Peng, Yameng, et al. "SWAP-NAS: Sample-Wise Activation Patterns For Ultra-Fast NAS." arXiv preprint arXiv:2403.04161 (2024).

---

> ### Author Response · Authors · 2025-01-09
> **Reply to Reviewer 6fJQ**
>
> ## **Newer proxy comparisons**
> Thank you for suggesting those experiments.
> We have added the SWAPNAS reference to the literature review.
> In Section 2.2 we added a sentence
> > Similarly, Peng et al. proposes to count the number of unique ReLU activation patterns across the data dimension.
>
> We have performed evolutionary search experiments with MeCo and SWAPNAS.
> As is evident from the table, our approach is compatible with these newer proxies.
> Following the suggestion of Reviewer pja4, we performed a statistical significance test for every pair (Avg vs. Stat) of the results, demonstrating that statistical approach outperforms averaging in most cases and in a number of instances it demonstrates a statistically significant advantage.
> | Search Space | Search Alg. | SWAPNAS |  |  MeCo | |
> |---|---|:---:|:---:|:---:|:---:|
> |  |  | Avg | Stat | Avg | Stat |
> | NAS-Bench-101 (CIFAR-10) | REA | 93.05$\pm$0.12 | 93.37$\pm$0.11 | 93.13$\pm$0.54 | 93.52$\pm$0.44 |
> |  | FreeREA | 93.42$\pm$0.36 | 93.71$\pm$0.32 | 93.21$\pm$0.45 | 93.51$\pm$0.39 |
> | NAS-Bench-201 (CIFAR-100) | REA | 71.77$\pm$0.49 | 72.54$\pm$0.56 | 71.31$\pm$1.37 | 71.54$\pm$1.11 |
> |  | FreeREA | 71.70$\pm$1.02 | 72.81$\pm$0.98 | 71.86$\pm$0.87 | 71.86$\pm$0.99 |
> | NAS-Bench-201 (ImageNet16-120) | REA | 44.39$\pm$0.98 | 45.74$\pm$1.22 | 42.47$\pm$0.84 | 42.81$\pm$0.81 |
> |  | FreeREA | 44.75$\pm$1.15 | 45.89$\pm$1.01 | 42.88$\pm$0.79 | 43.23$\pm$0.85 |
> | TransNAS-Bench-101 (Jigsaw) | REA | 92.36$\pm$0.44 | 93.17$\pm$0.30 | 92.43$\pm$0.14 | 93.64$\pm$0.13 |
> |  | FreeREA | 92.29$\pm$0.18 | 93.04$\pm$0.21 | 91.68$\pm$0.20 | 93.55$\pm$0.09 |
> | TransNAS-Bench-101 (Class object) | REA | 45.96$\pm$0.11 | 46.80$\pm$0.12 | 46.73$\pm$0.09 | 46.99$\pm$0.08 |
> |  | FreeREA | 47.01$\pm$0.07 | 47.09$\pm$0.05 | 46.60$\pm$0.10 | 47.11$\pm$0.12 |
>
> ## **Insufficient analysis**
> We have now conducted additional experiments using MeCo, SWAPNAS, Synflow, and LogSynflow.
> This demonstrates the benefits of our approach.
> Let us consider evolutionary search cases.
> Currently, 13 pairs Avg vs. Stat are presented for each search space.
> | Search space | Average accuracy improvement | \# cases of improvement out of 13 |
> |---|---|---|
> | NAS-Bench-101 (CIFAR-10) | 0.22 | 10 |
> | NAS-Bench-201 (CIFAR-100) | 0.57 | 8 |
> | NAS-Bench-201 (ImageNet16-120) | 0.72 | 11 |
> | TransNAS-Bench-101 (Jigsaw) | 1.08 | 13 |
> | TransNAS-Bench-101 (Class object) | 0.46 | 11 |
>
> Following the suggestion of Reviewer pja4, we have updated the tables, highlighting in bold the best performance for every pair of Avg vs. Stat.
> We conducted a statistical analysis for each pair.
> All statistically significant improvements are underlined.
>
> As discussed in the response to the next concern, we have now moderated the claim and specified the architectural spaces and ranking functions where improvement is observed.
>
> ## **Reformulate the claims**
> We thank you for pointing out the ambiguity of this claim.
> This particular quote refers to the search algorithm.
> We show how the random search algorithm and the evolutionary search algorithm have improved their performance in various search spaces in Table 1 and Table 2.
> We have reformulated it to avoid such confusion.
>
> >...that this often enhances the performance of random and evolutionary search algorithms for zero-shot NAS for a variety of commonly-used ranking functions and architectural spaces.
>
> We have added a specific list of proxies and a list of search algorithms that we are using in the experiments to the last paragraph of the introduction.
> > Specifically, we select Eigenvalue score (Mellor et al., 2020), ReLU Hamming distance (Mellor et al., 2021) and condition number of NTK (Chen et al., 2021) to demonstrate how random search (Mellor et al., 2020) and evolutionary search (REA (Real et al., 2019) and FreeREA (Cavagnero et al., 2023a)) achieve better performance in the majority of cases (>80 percent), when enhanced with our statistical evaluation approach, for standard architectural search spaces including NAS-Bench-101, NAS-Bench-201, and TransNAS-Bench-101.

---

> ### Comment · Reviewer_6fJQ · 2025-01-18
>
> Thank you for your big efforts in resolving my concerns, along with the additional experiments! Its nice to see that the approach is compatible with newer proxies as well, making the overall approach more robust. Thank you also for reformulating the claims! I do not have any further comments.

---

### Review · Reviewer_HmzA · 2024-12-19

**Summary Of Contributions:**

- The authors provide a detailed overview of previous work in the zero-shot NAS field. They formalize work in the field as the development of new search algorithms and ranking functions which can be paired to find the best candidates within an architectural search space. I found their exposition to be thorough and easy to follow. They argue that variability in the ranking functions due to evaluation on a randomly chosen minibatch is a pervasive issue that will impact results across the field.
- The authors then demonstrate that many commonly used ranking functions demonstrate significant levels of stochasticity as a function of the minibatch they are evaluated on. Furthermore, they show that 1) this variability appears to be a feature of each ranking function that is fairly stable across different architectural search spaces and 2) in general, within a search space, ranking functions exhibit lower variability on more performant architectures.
- The authors then describe a drop-in method to integrate awareness of ranking function variability into search algorithms. In particular, they suggest using a Mann-Whitney U test to test for stochastic dominance of ranking function evaluations from one architecture vs. another. With random tie-breaking, this method easily integrates into a variety of existing search algorithms.
- For random and evolutionary search, using this hypothesis testing framework generally leads to the selection of more performant architectures, regardless of the underlying ranking function or search algorithm.
- The authors also observe that in the evolutionary setting, caching of ranking function score samples (as opposed to reevaluation of previously seen architectures) leads to better performance.

**Audience:**

Yes

**Broader Impact Concerns:**

I do not have concerns on the ethical implications of the work that would require discussion in a Broader Impact Statement.

**Claims And Evidence:**

Yes

**Requested Changes:**

## Changes critical for securing my acceptance recommendation.
- Neither SynFlow and LogSynFlow as presented in Abdelfattah et al. 2021 and Cavagnero et al. 2023a use randomly generated data as input. Instead, these metrics calculate gradients relative to constant unit input. I have two requested changes with respect to this point:
	- Please correct the corresponding descriptions of these algorithms in the third paragraph of the introduction.
	- SynFlow is competitive with the eigenvalue score on several of the benchmarks evaluated in Abdulfattah et al. 2021. To better contextualize your results, I believe it would be important to consider how architectures selected via SynFlow (i.e. a non-stochastic ranking function) compare to architectures selected by stochastic ranking functions with the Mann-Whitney U test, as considered here.

## Changes that would strengthen this work.

- One of the biggest take-home messages of this work is that there is significant variability in ranking function evaluation which can meaningful influence the the outcome of zero-shot NAS procedures. This finding suggests that zero-shot NAS may be fruitfully considered as a stochastic optimization problem. Evaluating ranking function evaluations using a Mann-Whitney U test has been shown to be an method to handle this variability, but one could also imagine others performing well based on the properties of the noise (e.g. reinforcement learning, bayesian optimization, etc.) I believe a discussion of this point more generally would be useful to contextualize the findings of the paper, especially as some of these other methods have been previously applied to zero-shot NAS.

- It would be interesting to know if there is any relationship between 1) variability in the ranking function across minibatches and 2) variability in trained model performance (e.g. if one were to use those same minibatches as "test" data). Concretely I am asking:
	- For a fixed architecture, across the same set of minibatches, how correlated is the variability between different ranking functions?
	- If you were to then train this fixed architecture (excluding the evaluated minibatches from the training set), how would test performance on minibatches correlate with the variability of each ranking function?

If ranking function variability reflects something about variability in model performance, it may be possible to design zero-shot NAS strategies which use this varibiability to target objectives beyond average case performance. Such a study may also relate to the observation that better performing architectures within a search space exhibit less variability in their ranking functions.

- Please clarify how the Relu Hamming Distance differs from eigenvalue score. If both are derived from Mellor et al. 2021 I am unclear on how the Relu Hamming Distance maps onto the material described there.

- A subtle but interesting ablation would be to consider how the form of the ranking function comparison (i.e. comparison of means vs. Mann Whitney U statistic) influences architecture choice, independently of statistical significance testing. For example either increasing the threshold (i.e. more lenient testing criteria) of the Mann Whitney U test, or performing a t-test instead of simple average comparison would be interesting to consider in addition to the evaluation criteria studied here.

**Strengths And Weaknesses:**

# Strengths
- To my understanding, the characterization of ranking function variability within and across many architectural search spaces is a novel and useful contribution to the field. It is interesting to observe that ranking functions generally become less variable on better performing architectures within a search space, and this is a phenomena which would be interesting to describe to the community.
- The benchmark evaluation results demonstrating the impact of integrating statistical testing into search algorithms are interesting. These results demonstrate that variability in the ranking function does appear to have a meaningful impact on the performance of many zero-shot NAS setups, and the authors essentially describe a method of improving zero-shot NAS which is is orthogonal to existing methods which seek to develop improved ranking functions or search algorithms.
- Finally, I found the description of the literature in the field to be fairly thorugh and well suited to contextualize the results here.

# Weaknesses
- The main weakness I observe in the work is an omission of the role of non-stochastic ranking functions in zero-shot NAS. I have requested changes related to this point below.
- Beyond this point, I believe the work could benefit from some minor points of clarification around the methods used as well as a more thorough discussion of implications of this work for the field. I have described my requested changes corresponding to this point below.

---

> ### Author Response · Authors · 2025-01-09
> **Reply to Reviewer HmzA Part 1**
>
> ## **SynFlow and LogSynFlow**
> Thank you for pointing this out.
> Yes, SynFlow uses a unit vector input and we have corrected the third paragraph of the introduction with the following text:
>
> > There are several data-agnostic ranking functions, including SynFlow and LogSynFlow, which use a batch of {\em constant unit inputs}, and Zen-Score, which uses a batch of {\em randomly generated} data as an input.
>
> We have added evolutionary search experiments with LogSynFlow and Synflow in the Appendix.
> See Table 4.
> These particular ranking functions are not well-suited to our stochastic method, and averaging yields higher performance for them.
> We hypothesise that there might be cases where a set of architecture scores is sampled in a way such that stochastic dominance cannot be identified, which essentially collapses STAT-MAX from Algorithm 1 to a random choice.
>
>
> | Search Space                                        | Search Alg. |     SynFlow    |                |   LogSynFlow   |                |
> |-----------------------------------------------------|-------------|:--------------:|:--------------:|:--------------:|:--------------:|
> |                                                     |             |       Avg      |      Stat      |       Avg      |      Stat      |
> | NAS-Bench-101 (CIFAR-10)          | REA         | 93.69$\pm$0.38 | 93.56$\pm$0.31 | 93.76$\pm$0.36 | 93.62$\pm$0.28 |
> |                                                     | FreeREA     | 93.80$\pm$0.24 | 93.74$\pm$0.32 | 93.76$\pm$0.25 | 93.77$\pm$0.36 |
> | NAS-Bench-201 (CIFAR-100)     | REA         | 71.52$\pm$0.50 | 71.34$\pm$0.47 | 71.94$\pm$0.50 | 71.53$\pm$0.62 |
> |                                                     | FreeREA     | 71.62$\pm$0.73 | 72.46$\pm$1.04 | 72.23$\pm$0.68 | 73.14$\pm$0.39 |
> | NAS-Bench-201 (ImageNet16-120)    | REA         | 44.29$\pm$1.02 | 44.57$\pm$0.99 | 45.03$\pm$0.82 | 44.91$\pm$1.04 |
> |                                                     | FreeREA     | 44.98$\pm$0.89 | 44.74$\pm$1.21 | 45.06$\pm$0.90 | 45.16$\pm$0.95 |
> | TransNAS-Bench-101 (Jigsaw)       | REA         | 91.84$\pm$0.17 | 92.71$\pm$0.41 | 91.92$\pm$0.33 | 92.49$\pm$0.28 |
> |                                                     | FreeREA     | 91.46$\pm$0.25 | 92.22$\pm$0.53 | 91.46$\pm$0.20 | 91.89$\pm$0.62 |
> | TransNAS-Bench-101 (Class object) | REA         | 46.51$\pm$0.03 | 46.82$\pm$0.10 | 46.50$\pm$0.06 | 46.82$\pm$0.10 |
> |                                                     | FreeREA     | 46.75$\pm$0.07 | 47.12$\pm$0.05 | 46.66$\pm$0.04 | 47.03$\pm$0.09 |
>
> ## **Stochastic optimisation problem**
> Thank you for this suggestion.
> The idea of zero-shot NAS as a stochastic optimisation problem is interesting.
> We added the following paragraph to the conclusion section.
>
> >Every ranking function evaluation for a given architecture can be considered equivalent to sampling from the distribution.
> In our work, we handled such sampling with statistical testing in order to retain compatibility with the existing pipelines and search agorithms.
> Formulating a zero-shot neural architecture search as a stochastic optimisation problem that interprets the output of a ranking function not as a score, but as a distribution could be a potential research direction.
>
> ## **For a fixed architecture, across the same set of minibatches, how correlated is the variability between different ranking functions?**
> We are unable to perform this experiment in the exact requested setting due to the way our data were collected and the inability to recollect them within a rebuttal timeframe.
> Specifically, we have collected and cached 20 evaluations for each ranking function in each architecture of each search space.
> Since those evaluations are the most computational-heavy operations, it allowed to make all subsequent experiments rather quickly, using cached scores.
> First, this collection uses random minibatches, not the same for different ranking functions.
> This is still fine for computing the correlation, just less clear than in the suggested setting.
> Second, 20 evaluations are not enough for the experiment you have proposed.
> Let us assume that variation is still computed over 10 evaluations, as in the other experiments in our paper.
> For clear results, we need minibatches to be different.
> It means that on our cached data, we have only 2 datapoints of variations per ranking function.
> Two sets of two points each are not enough to compute the correlation between them.
> Third, it is unclear what such correlation is meant to demonstrate?
> This is important to answer beforehand given the amount of work this experiment requires.
> So, how can knowledge that variation of a ranking function A highly correlates with variation of a ranking function B be utilised?

---

> > ### Author Response · Authors · 2025-01-09
> > **Reply to Reviewer HmzA Part 2**
> >
> > ## **If you were to then train this fixed architecture (excluding the evaluated minibatches from the training set), how would test performance on minibatches correlate with the variability of each ranking function?**
> > If understand this request correctly, Figure 3 illustrates the relationship between the test accuracy of a trained network and the variability of each ranking function.
> > Although it employs the coefficient of variation rather than the variation itself, it should meet your requirements.
> >
> > ## **How the Relu Hamming Distance differs from Eigenvalue score**
> > Mellor et al. introduced the eigenvalue score in an earlier arXiv version of their paper, and the ICML version of the paper then presented results using the ReLU Hamming distance. In the revised version of our paper, we now cite the earlier arXiv version separately (with the provided URL) when referring to the Eigenvalue score.
> >
> > Our paper has the sentence
> > > Based on the Neural Tangent Kernel (NTK) (Jacot et al., 2018), the Eigenvalue score uses the eigenvalues of the gradient correlation matrix to predict the learning capabilities of an architecture'' to describe how the eigenvalue score is constructed.
> >
> > We have added the following sentence into the literature review to explain the construction of the ReLU Hamming distance:
> > >The ReLU Hamming distance is constructed as a determinant of the kernel of Hamming distances betweent activation patterns of ReLU, where a pattern is represented as a string of zeros and ones.
> >
> > The Eigenvalue score is based on the eigenvalues of the gradient correlation matrix; by contrast, the ReLU Hamming distance is constructed based on the determinant of a matrix whose entries are derived from the Hamming distance between the ReLU activation patterns for different inputs in the batch.
> >
> > ReLU Hamming Distance has variation similar to the other ranking functions, because the ReLU activation pattern depends on both the initialisation of the weights and the selected input batch.
> > Therefore, the ReLU Hamming distance fits into the framework presented in our paper.
> >
> > ## **Mann Whitney U test threshold**
> > Thank you for this suggestion.
> > We have added an ablation on the threshold for the STAT-MAX procedure to the Appendix.
> > The optimal threshold lies within the range from 0.025 to 0.075.
> > The lower threshold seems to be excessively stringent, resulting in reduced performance.
> > Conversely, the higher threshold is overly lenient, permitting false cases of stochastic dominance.

---

> > > ### Comment · Reviewer_HmzA · 2025-01-11
> > > **Thank you for your response.**
> > >
> > > Thank you for your response. I have included futher correspondence organized by section headings.
> > >
> > > # Synflow and LogSynFlow
> > > - Thank you for including these results. It is interesting that SynFlow and LogSynFlow are not improved by the stochastic method; I believe this result is an interesting ablation study in its own right.
> > > - However, I think it is more critical to note that the performance of averaging under SynFlow/LogSynFlow appears competitive with the results of applying your stochastic method as presented in Table 3. I do not think that this result weakens the conclusions of the paper, but I believe that it is important to clearly discuss a comparison to these non-stochastic methods in the main text to provide more context for your results.
> > >
> > > # Stochastic optimization Problem.
> > > - I appreciate you making this addition.
> > >
> > > # For a fixed architecture, across the same set of minibatches, how correlated is the variability between different ranking functions?/If you were to then train this fixed architecture (excluding the evaluated minibatches from the training set), how would test performance on minibatches correlate with the variability of each ranking function?
> > >
> > > I believe there is some misunderstanding here. I will respond for the sake of clarity, but I don't believe there to be any issue with the experimental methodology currently described in the paper, and these points do not require further follow up from the authors. I am just curious about ranking function variability.
> > > - One could imagine that minibatch A of data contains "harder" examples than minibatch B, just based on random chance.
> > > - Given ranking functions $f$ and $g$, is it the case that $\mathbb{E}[f(A)]>\mathbb{E}[f(B)]$, averaging over stochasticity in network optimization/initialization? How about for ranking function $g$? In other words, does the variability of any (or all) ranking function(s) correlate with some sort of variability across minibatches (e.g. sample difficulty, class distribution of samples...)?
> > > - Discovering such a correlate for ranking function variability would be interesting in its own right, but it would also suggest that the there could be value to selecting the particular set of minibatches used for zero-shot NAS.
> > >
> > > # How the Relu Hamming Distance differs from Eigenvalue score
> > > - Thank you for these clarifications!
> > >
> > > # Mann Whitney U test threshold
> > > - Thank you for these insights.

---

> > > > ### Author Response · Authors · 2025-01-20
> > > > **Response for additional comments**
> > > >
> > > > Thank you for your clarification on the selection of minibatches.
> > > > In fact, this is an interesting research direction that is very close to the methodology described in this paper.
> > > > The research you are suggesting just requires to run our methodology for longer, which we never accounted for during the data collection in our experiments.
> > > > Practically, one could optimise both a minibatch and a seed in that way.
> > > > This poses an interesting methodological question of whether any ranking function could be presented as the best on a particular search space, with a careful selection of the seed and the set of minibatches.
> > > >
> > > > We added some results for SWAPNAS, MeCo, and Synflow to Section 5.3.
> > > > They form a new table (Table 3) that covers NAS-Bench-101 (CIFAR-10) and NAS-Bench-201 (ImageNet16-120) search spaces.
> > > > Those results are referred to in the newly added paragraph:
> > > >
> > > > > In Table 3 we demonstrate that our approach is generalisable to the newer ranking functions. SWAPNAS (Peng et al., 2024) and MeCo (Jiang et al., 2024) are recently published Zero-Shot ranking functions
> > > > that aim to achieve the highest correlation with the accuracy of the architecture in the NAS-Bench search
> > > > spaces. Table 3 shows that the proposed methodology can easily be adapted to the new ranking functions.
> > > > In addition to that, we provide results on data-agnostic SynFlow (Abdelfattah et al., 2021) and LogSynFlow (Cavagnero et al., 2023a). Unlike any other ranking functions presented in this paper, the variation
> > > > of SynFlow and LogSynFlow is induced only by random initialisation of the evaluated architecture. Those
> > > > data-agnostic approaches present a challenge to our statistical evaluation. Averaging demonstrates performance comparable or higher to that of our statistical evaluation on them. More results for SWAPNAS, MeCo, SynFlow and LogSynFlow are demonstrated in the Appendix C.
> > > >
> > > > The following paragraph is added to the conclusion.
> > > >
> > > > > We have observed that the statistical method does not consistently benefitial for data-agnostic SynFlow and LogSynFlow, while the their performance is competitive with other ranking functions discussed in the paper.
> > > > This suggests that an alternative strategy to the statistical approach is to develop techniques that are minimally affected by the data batch and initialisation.

---

### Review · Reviewer_5g6C · 2024-12-20

**Summary Of Contributions:**

The paper focuses on mitigating the randomness of NAS Zero-shot ranking methods. To enable noise robust evaluation of architecture with the zero-shot metrics, the paper propose to use statistical TOP-K method where Mann–Whitney U test is used to identify statistically better architectures. The author conducted experiments using random and evolutionary searches to justify the effectiveness of their method.

**Audience:**

Yes

**Claims And Evidence:**

No

**Requested Changes:**

See weakness above.

**Strengths And Weaknesses:**

Strengths:
* The paper is well written, and the motivation is clear and easy to follow.
* Statistical MAX/TOP-K is a generalized method for noisy proxy measurement, so potentially a border community might be interested in applying this trick.
* The author did comprehensive experiments with consistent results, showing the effect of mitigating noise with statistical methods.

Weakness:
* The notation used in the paper is not consistent which leads to some confusion that I e.g.:
(a) In algorithm 1, K refers to top K architectures while in section 4.1 K seems to refer to the number of batches
(b) In algorithm 1, N refers to how much ranking function measurements, however in Table 1 N refers to how much search budget?
can the author clarify these and fix the consistency of the notation?

* The improvement is quite marginal for all zero-cost-proxy: almost all improvements are within 1 standard deviation. Why is this the case, can the author shed some light on this?

* In all the experiments, the author seems to use the same number of ranking function evaluations for the architecture (10 times). How would the performance of average estimation and Stat estimation change with different evaluation budgets? I guess this is to answer the important practical question: how much budget of zero-poxy evaluation do I need if I estimate the stochastic dominance?

* The author selected 3 zero-cost metrics and evaluated them on 8 datasets, however, it's not clear to me which metric benefits the most from statistical estimation compared to average estimation. Is it bringing more improvement to weaker or more noisy zero-shot metrics? I t would be nice if the author can show straightforward relative improvement between stat and avg for different rank functions (e.g. showed in Figure 3)

---

> ### Author Response · Authors · 2025-01-09
> **Reply to Reviewer 5g6C**
>
> **W1**: Thank you for pointing this out. Since "Top-K" is a standard and widely-used term, we have replaced the number of evaluations $K$ with $V$ in equation (1).
> We have modified the notation of Algorithm 1 to ensure that it does not use N, thus avoiding confusion.
> The caption in Table 1 starts with a clear indication of the meaning of N. To further eliminate potential confusion, the notation of Equation (1) has been changed from $N$ to $N_{SS}$ for the size of the search space.
>
> **W2**: In almost all cases, our approach improves the results compared to averaging, even if the improvement is not as substantial.
> For example, let us consider evolutionary search cases.
> Currently, 13 pairs Avg vs. Stat are presented for each search space.
>
> | Search space | Average accuracy improvement  |  # cases of improvement out of 13 |
> | --- | --- | ---|
> | NAS-Bench-101 (CIFAR-10)          | 0.22 | 10  |
> | NAS-Bench-201 (CIFAR-100)         | 0.57 | 8   |
> | NAS-Bench-201 (ImageNet16-120)     | 0.72 | 11  |
> | TransNAS-Bench-101 (Jigsaw)        | 1.08  | 13  |
> | TransNAS-Bench-101 (Class object)  | 0.46  | 11 |
>
> Our work is the first to consider the randomness of ranking functions. The novel approach is relatively simple, and is also specifically designed to work on top of the existing methodology. It is already common practice to evaluate a ranking function several times. Given that there is no real additional computational cost, a relatively consistent improvement, even if small, is a welcome contribution.
>
> Following the suggestion of Reviewer pja4, we have updated the tables, highlighting in bold the best performance for every pair of Avg vs. Stat.
> We conducted a statistical analysis for each pair.
> All statistically significant improvements are underlined. In multiple cases, our approach demonstrates statistical significance.
>
> **W3**:
> In our experiments, 10 evaluations were deliberately chosen as a trade-off between computational overhead and statistical test performance.
> The evaluation of the ranking function is the most costly operation in any NAS search algorithm, including ours, so it would be preferable to have as few evaluations as possible.
> By contrast, statistical tests are more precise for a larger number of samples, so it would be preferable to have as many samples as possible.
>
> Overall, in our experiments, 6 to 15 evaluations are sufficient to estimate stochastic dominance with the Mann-Whitney U-test. We recommend using 8-10 as we empirically find it to be a good trade-off for most of the ranking functions, however, our ablation for this number is limited.
> More than 10 slows down the search when compute-heavy ranking functions are used (like any based on the NTK).
> Fewer than 8 reduces the reliability of the test.
>
> **W4**: Our experiments do not expose a significant difference between the improvement for low- versus high-variance ranking functions.
> In our experiments with evolutionary search, Eigenvalue score (low variance) has the average improvement of 0.73, whereas for RELU dist (high variance) the average improvement is 0.60.
> This clarification has been added to Section 5.3.

---

### Decision · Action_Editor_Chzm · 2025-01-28

**Recommendation:** Accept as is

**Comment:**

The reviewers mostly reached an agreement recommending acceptance of the paper and believed that most of the concerns have been sufficiently addressed in the revisions and authors replies. As mentioned above, there were some concerns on the applicability of the method since it addresses a rather specific problem (randomness of metrics in zero-shot NAS) -- while the AE believes that the paper meets the two TMLR criteria (soundness and audience) and thus recommends acceptance as-is, the authors are still encouraged to at least add some qualitative discussions (e.g., similar to their reply to Reviewer 5g6C) about this point in their camera-ready version which I believe would also increase the impact of their work.

**Audience:**

Overall, the reviewers agree that the work will be of interest to the NAS / AutoML community. There were some concerns raised that whether the method only works effectively in certain search spaces but not convenet-based benchmark spaces (which led to concerns on broad applicability of the method), but overall the AE believes that the work has successfully established that it is of interest to at least some members of the TMLR community to satisfy this criterion.

**Claims And Evidence:**

This paper aims to address the issue in zero-shot NAS that the outputs of the ranking functions (which scores architecture candidates without training them) can vary significantly and the common approach of averaging across the different functions is unprincipled. The authors propose a ranking function variance analysis framework and present both an analysis of ranking function and shows that its variance is useful for enhancing performance of various zero-shot NAS algorithm.

Overall, the reviewers agree that (after several rounds of revision), all claims in the paper are well-supported by evidence and are technically sound. The AE agrees with this judgement, and believes this submission meets this bar.